# CLIPDraw++: Text-to-Sketch Synthesis with Simple Primitives

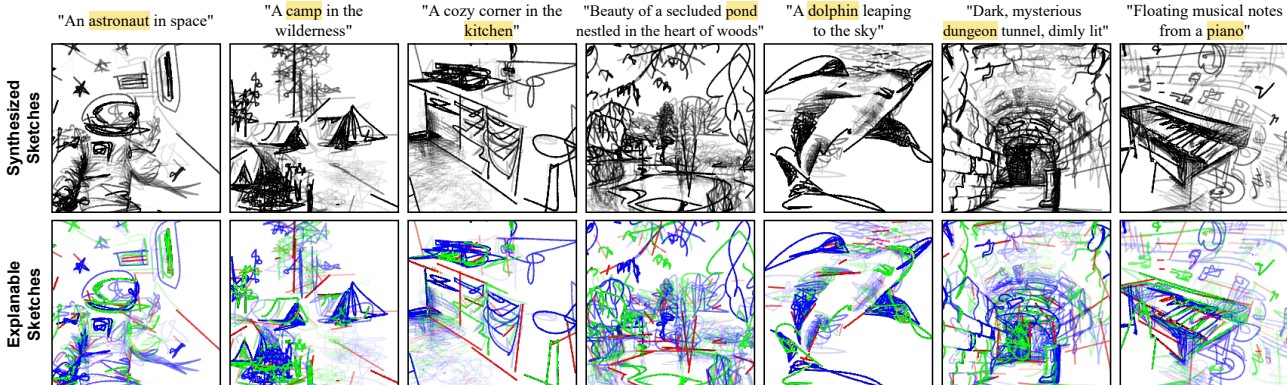

Figure 1. Our method, CLIPDraw++ synthesizes vector sketches conditioned on an input text prompt using simple primitive shapes like circles, straight lines, and semi-circles, with focus on the highlighted words.

## Abstract

*With the goal of understanding the visual concepts that CLIP associates with text prompts, we show that the latent space of CLIP can be visualized solely in terms of linear transformations on simple geometric primitives like straight lines and circles. Although existing approaches achieve this by sketch-synthesis-through-optimization, they do so on the space of higher order Bézier curves, which exhibit a wastefully large set of structures that they can evolve into, as most of them are non-essential for generating meaningful sketches. We present CLIPDraw++, an algorithm that provides significantly better visualizations for CLIP text embeddings, using only simple primitive shapes like straight lines and circles. This constrains the set of possible outputs to linear transformations on these primitives, thereby exhibiting an inherently simpler mathematical form. The synthesis process of CLIPDraw++ can be tracked end-to-end, with each visual concept being expressed exclusively in terms of primitives.*

## 1. Introduction

Simplified representations like sketches and verbal descriptions are potent mediums for communicating ideas, focusing on the core essence of the subject. While language conveys abstract meanings, sketches capture visual specifics. For instance, a designer might sketch a client's ideas for clarity during reviews of design plans, and automating this process could cut labour costs. Understanding the importance of this text-to-sketch generation task, several research initiatives have explored text-conditioned sketch generation, utilizing the CLIP model [24] and the transformative diffusion models [25]. CLIPDraw [5] creates drawings from text using pretrained CLIP text-image encoders, while VectorFusion [11] adapts text-guided models for vector graphics, without relying on extensive datasets. Such models are thus finding a growing role in art and design [7, 11], often surpassing human performance in many tasks [12, 26]. However, the complexity of the underlying algorithms tend to grow with with their performance, making them less transparent, and hence, less controllable.

To this end, we aim to cast the problem of sketch synthesis exclusively in terms of mathematically tractable primitives. The atoms of our algorithm consist of basic geometric shapes like straight lines, circles, and semi-circles, with a synthesis process that can be completely summarized as a linear transformation over these basic shapes. The benefit of such a construction is two-fold – (1) the number of parameters that need to be optimized is dramatically reduced, and (2) each step of the synthesis process can be clearly tracked and expressed via a closed form linear expression. Our approach is based on synthesis-through-optimization, that visualizes using vector strokes, concepts encoded in

the representation space of CLIP [24] corresponding to natural language text prompts. Such vector strokes, as emphasized by CLIPDraw [5], enable clearer breakdowns and simpler component attribution than pixel images. While advanced models like diffusion can produce high-quality pixel sketches [25], their pixel complexity can obscure underlying logic. However, using vectorized strokes, especially when limited to primitive shapes, enhances clarity and a better understanding of the steps that a model takes to synthesize a sketch, while significantly lowering the number of parameters required to achieve state-of-the-art results.

In this paper, we present CLIPDraw++, an algorithm that can synthesize high-quality vector sketches based on text descriptions. It not only synthesizes sketches but also tracks the evolution of each vector stroke from the initial geometric primitives, like straight lines, circles, and half-circles, to the final output. Leveraging the cross-attention maps from a pretrained text-to-image diffusion model [25], we define an initial sketch canvas as a set of primitive shapes, with their parameters tuned using a differentiable rasterizer [15]. In order to ensure minimality and a simple mathematical form, we have constrained the primitives to adhere to specific initial shapes with more rigid geometries than Beźier curves. Our CLIPDraw++, built upon CLIPDraw [5], differentiates itself by using predefined primitive shapes for strokes, emphasizing simplicity, while CLIPDraw initializes strokes with complex, arbitrarily-shaped Bézier curves, that exhibit a wastefully large set of shapes which they can transform into, but are not necessary for producing meaningful sketches. Our method linearly transforms these simple primitives into corresponding sketch strokes, unlike the more abstract and harder-to-understand sketches produced by CLIPDraw. Our CLIPDraw++ synthesizes superior sketches in a more efficient manner, both in terms of performance and memory usage, when compared to the existing methods. This improvement is attributed to the use of fewer primitives, each with fewer control points strategically distributed across the canvas. Unlike prevalent generative models [19], including diffusion models [25] that demand extensive parameter training, our CLIPDraw++ synthesizes sketches via optimization and operates without any specific training. Instead, a pretrained CLIP model is used as a metric for maximizing similarity between the input text prompt and the synthesized sketch.

In summary, we make the following contributions – **(1)** Pose the problem of sketch synthesis via optimization in terms of a well understood mathematical framework of learning linear transformations on simple geometric primitives; **(2)** Propose a sketch canvas initialization approach using primitive shapes. By leveraging the cross-attention map of a pre-trained diffusion model, we strategically distribute these primitives across the canvas based on necessity. Furthermore, by initializing these primitives with a low opac-

ity, the model accentuates only those primitives pertinent to the text prompt; **(3)** Primitive-level dropout as an innovative technique to "regularize" our optimization. By doing so, we effectively diminish over-optimization, cut down on noisy strokes, and elevate the overall quality of the synthesized sketches; **(4)** Extensive qualitative and quantitative experiments demonstrate the usefulness of our novel components in delivering performance surpassing existing methods, while exhibiting a significantly simpler and parameter-efficient synthesis scheme.

## 2. Related Works

**Sketch Generation:** Free-hand sketches communicate abstract ideas leveraging the minimalism of human visual perception. They aim for abstract representations based on both structural [3] and semantic [31] interpretations. Digital sketching methods aiming to mimic human drawing span a wide range of representations, from those founded on the input image's edge map [4, 14, 16, 17, 29] to ones that venture into a higher degree of abstraction [2, 5, 8], typically represented in vector format. In the realm of vector sketch creation, initiatives such as CLIPasso [31] and CLIPascene [30] are dependent on an image input; meanwhile, approaches like CLIPDraw [5], VectorFusion [11] and DiffSketcher [32] engage with text-based conditional inputs, standing distinct from other unconditional methodologies. Specifically, CLIPDraw [5] employs optimization-based sketch synthesis, whereas both VectorFusion [11] and DiffSketcher [32] adopt diffusion-based approaches, with VectorFusion focusing on optimizing in the latent space, making it more light-weight and efficient compared to DiffSketcher that performs its optimization in the image space. In this paper, we explore text-conditioned sketch synthesis. Distinct from the existing approaches, we show that complex sketches can be synthesized as simple linear transformations on simple geometric primitives, with our novelty lying in the minimalism of our formulation.

**Synthesis through Optimization:** Instead of directly training a network to generate images, a different strategy called *activation maximization* optimizes a random image to match a target during evaluation [21]. CLIPDraw [5] advances this approach by using a CLIP language-image encoder [6, 24] to lessen the disparity between the synthesized image and a specified description, focusing on broad features rather than fine details. Although synthesis through optimization often results in unnatural or misleading images [20], employing 'natural image priors' can maintain authenticity [21, 22], which often involve the restrictive and computationally intensive use of GANs. In this paper, we extend the capabilities of CLIPDraw to synthesize sketches of objects and scenes through geometric transformations of primitive shapes, contrasting from CLIPDraw's approach, which utilizes complex and abstract sketches created with

hard-to-analyze, arbitrarily-shaped Bézier curves.

**Vector Graphics:** We leverage the differentiable renderer for vector graphics pioneered by [15], a tool no longer confined to vector-specific datasets thanks to recent advancements. The advent of CLIP [24], which fosters improved visual text embedding, has spurred the development of robust sketch synthesis techniques including CLIPDraw [5], CLIP-CLOP [18], and CLIPascene [30]. The recently introduced VectorFusion [11] also integrates a differentiable renderer with a diffusion model, aiding in the production of vector graphics creations like iconography and pixel art.

**Simplified representations for sketches:** Current sketch research on making neural sketch representations transparent and simplified is significantly limited, primarily concentrating on interpreting human sketches through stroke-level abstraction [1] and stroke location inversion [23]. However, as generative AI finds increasingly growing use in content creation, breaking down the abstraction inherent in such models for sketches [5, 30, 31] becomes ever more important. We present an approach that not only synthesizes sketches in terms of mathematically simple primitives, but also provides human-understandable insights into the representation space of foundation models like CLIP [24].

## 3. CLIPDraw++

In this work, we aim to synthesize sketches by expressing them as a collection of linearly transformed primitives, which could be evolved from an initial canvas into a set of strokes in a final sketch in a trackable manner based on an input text. We begin by populating a canvas with primitive shapes like straight lines, circles, and half-circles, detailed in Sec. 3.2. We then track their progress during the sketch synthesis process outlined in Sec. 3.3, guided by the training criteria specified in Sec. 3.4. An illustration of the proposed approach is provided in Fig. 2.

### 3.1. Sketch Synthesis from Primitives

Given an initial canvas $\mathbb{C}$ composed of a set of primitives $\{p_1, p_2, ..., p_n\}$ as $\mathbb{C} = p_1 \oplus p_2... \oplus p_n$, we qualitatively show that any semantically meaningful sketch $\mathbb{Y}$ can be constructed as a transformation $f(\cdot)$ on $\mathbb{C}$ such that $\mathbb{S} = f(\mathbb{C}) = f_1(p_1) \oplus f_2(p_2)... \oplus f_n(p_n)$, where $f_1, f_2, ..., f_n$ are linear transformations on the primitives, and $\oplus$ denotes the composition of the primitives into a single canvas. In other words, any primitive $p \in \mathbb{C}$ maps to a concept $y \in \mathbb{Y}$, the target sketch as $y = F \cdot p$, where $F$ is a matrix encoding the linear transformation. The learning problem thus becomes finding the best approximation $\tilde{F}$ of $F$ such that $p$ is appropriately transformed to depict some target concept in $\mathbb{Y}$. Thus, given an initial canvas and a target sketch, across all sub-concepts $y_i \in \mathbb{Y}$ in the target, the following objective needs to be optimized:

$$\min_{\mathcal{F}} l(\mathbb{C}_{\mathcal{F}}, \mathbb{Y}) = \arg\min_{i,\mathcal{F}} \sum_j ||y_i - \tilde{F} \cdot p_j|| \quad (1)$$

where $\mathcal{F} = \{\tilde{F}_1, \tilde{F}_2, ..., \tilde{F}_n\}$ is a set of linear transformations applied on $\{p_1, p_2, ..., p_n\}$ from $\mathbb{C}$ respectively to obtain an approximation of the target sketch $\mathbb{Y}$, and $\mathbb{C}_{\mathcal{F}}$ denotes the sketch obtained under the set of transformations $\mathcal{F}$ applied on $\mathbb{C}$.

However, the target concept sketch $\mathbb{Y}$ is not available, since the main objective is to synthesize it from a textual description. We thus choose a vision-language model, specifically, CLIP [24], as a proxy for the target sketch. The representation space of CLIP is unified, *i.e.*, natural language sentences and their corresponding visual counterparts have the same embedding. So, a sketch that captures the same semantics conveyed through a text prompt should have the same embedding in the representation space of CLIP. We thus make the CLIP text and sketch embeddings act as proxy representations for the ground-truth concept $\mathbb{Y}$ sketch and the synthesized sketch $\mathbb{C}_{\mathcal{F}}$ respectively and optimize the following objective which is equivalent to Eq. (1):

$$\min_{\mathcal{F}} l(\mathbb{C}_{\mathcal{F}}, x_t) = \arg\min_{\mathcal{F}} \left\| \mathcal{T}(x_t) - \mathcal{I}\left(\bigcup_i \tilde{F}_i \cdot p_i\right) \right\| \quad (2)$$

Thus in the formulation of Eq. (2), the text prompt $x_t$ serves as a proxy for the ground truth $\mathbb{Y}$.

### 3.2. Primitive-based Canvas Initialization

We initialize the sketch canvas by first identifying the essential landmark points derived from the textual input. Subsequently, these landmarks are filled using primitive shapes like straight lines, circles, and half-circles.

**Identifying Landmarks using Attention Maps:** CLIP-Draw++ initializes a canvas using primitives, which requires identifying important landmarks on that canvas based on the text input. For that, we employ the DAAM [28] generated token wise cross-attention mechanism as an integral component derived from the UNet architecture from latent diffusion model [25]. Specifically, we capture attention features separately from both upper and lower sample blocks, merging them into a unified attention map. This composite attention map is then normalized using the softmax function to create a probabilistic distribution map. This distribution map serves as the basis for our subsequent task of selecting $k$ positions. The magnitude of each point within this attention-derived distribution map is leveraged as a weight parameter to guide the selection process. This ensures that the selection of positions is influenced by the saliency and significance of the underlying attention-based features. To improve convergence towards semantic depictions, we place the primitive shapes or the initial strokes based on the salient regions (here patch) of the target image.

**Patch-based Initialization:** Point-based initialization is effective for similar stroke types [5, 31], but determining the

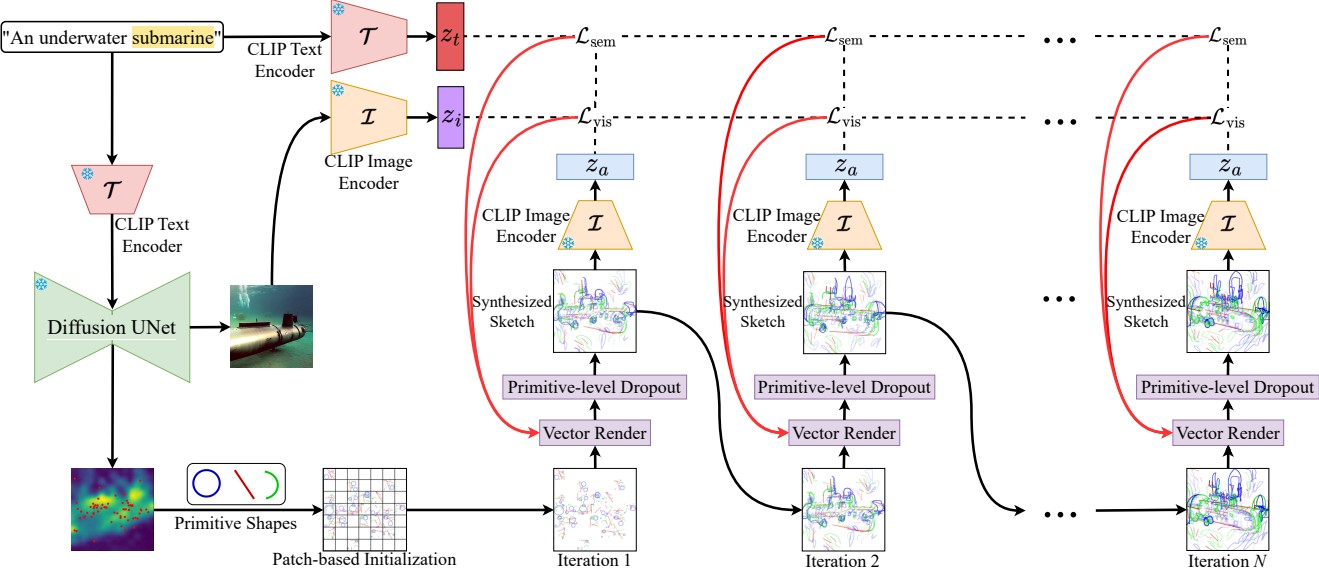

Figure 2. CLIPDraw++ comprises strategic canvas initialization, which utilizes diffusion-based cross-attention maps and a patch-wise arrangement of primitives, along with a primitive-level dropout (PLD). The proposed model, coupled with the use of a pre-trained image ($\mathcal{I}$) and text ($\mathcal{T}$) encoders from the CLIP model for similarity maximization, positions itself as an efficient and user-friendly tool in the realm of AI-driven sketch synthesis. The highlighted word is used to create the cross-attention maps.

right primitive for a specific point on the attention map can be difficult. Additionally, placing different primitives at a single location is problematic as it may result in clutter due to high point density, leading to uneven primitive distribution and messy sketches. To address this, our CLIPDraw++ introduces primitives in fixed ranges or 'patches', representing all points within, rather than at precise attention map locations. It divides a $224 \times 224$ canvas into patches of $32 \times 32$. Each patch receives a mix of basic primitives: straight lines, circles, and semi-circles, promoting uniform shape distribution around the attention map local maxima. The benefits of patch-based over point-based initialization are discussed in Sec. 4.3 and illustrated in Fig. 7.

**Initializing Sketch Canvas with Primitives:** We define a sketch as a set of $n$ strokes $\{s_1, \ldots, s_n\}$ appearing in a canvas. In order to elucidate the origins and evolution of these strokes, we initialize our canvas with primitive shapes, such as straight lines, circles, and semi-circles. Each primitive shape is created using a two-dimensional shape that employs two to four control points, represented as $s_i = \{p_i^j\}_{j=1}^c = \{(x_i, y_i)^j\}_{j=1}^c$ and an opacity attribute $\alpha_i$; where $c \in \{2, 3, 4\}$ denotes the number of control points. For example, straight line has 2 control points, while semi-circle and circle have 3 and 4. We incorporate the position of each control point and opacity of the strokes into the optimization process and use the semantic knowledge in CLIP to guide the synthesis of a sketch from a textual description. The parameters of the strokes are fed to a differentiable rasterizer $\mathcal{R}$, which forms the raster sketch $\mathcal{S} = \mathcal{R}((s_1, \alpha_1), \ldots, (s_n, \alpha_n)) = \mathcal{R}((\{p_1^j\}_{j=1}^c, \alpha_1), \ldots, (\{p_1^j\}_{j=1}^c, \alpha_n))$.

### 3.3. Optimizing Sketch Synthesis

Unwanted strokes can introduce noise into a sketch, making the removal of unnecessary strokes vital for automated sketch creation. This section outlines procedures for eliminating noisy strokes, drawing inspiration from existing machine learning techniques and human sketching practices.

**Primitive-level Dropout:** Dropout [27] is a regularization technique for neural networks where random subsets of neurons are temporarily deactivated during training. This procedure reduces overfitting by preventing co-adaptation of feature detectors and promoting a more robust network representation. Inspired by the success of dropout in learning robust representation, we propose *primitive-level dropout* (PLD) in our CLIPDraw++ model. This technique focuses on optimizing the use of each sketch primitive and removing any that contribute unnecessary noise. The intuition behind our approach is that limiting the number of available primitives compels the model to efficiently use each one to capture the semantics described in the texts, avoiding their wastage in noisy strokes. By selecting random smaller subsets of primitives for each iteration, the model is encouraged to utilize every primitive across all iterations for meaningful sketch representation, thereby minimizing their use in creating unnecessary noise.

In each step of the optimization process, a specific number of primitives, represented by $\mathcal{P}$ and determined through a probability distribution, are intentionally removed, after which a gradient step is undertaken. Formally, a random subset of the original primitives $\mathbb{C}$ is selected in each iteration to create a reduced canvas $\tilde{\mathbb{C}}$ as follows:

$$\mathbf{d} \sim \text{Bernoulli}(1 - \mathcal{P}); \quad \tilde{\mathbb{C}} = \mathbb{C} \cdot \mathbf{d}^T,$$

where $\mathbf{d}$ is an $n$-dimensional row vector of Bernoulli random variables, each of whose elements are 1 with probability $(1 - \mathcal{P})$, and 0 otherwise. Multiplying $\mathbf{d}$ with $\mathbb{C}$ masks out the primitives whose indices correspond to the elements of $\mathbf{d}$ that are 0, while the others are retained. Subsequently, these $\mathcal{P}$ primitives are reintroduced into the optimization loop at the conclusion of the iteration, with another random subset of $\mathcal{P}$ primitives being removed in the next iteration. The cyclical approach of introducing, excluding, and then reintroducing primitives in CLIPDraw++ offers a balanced optimization, ensuring sketches are not overwhelmed with strokes but still retain vital elements. Without loss of generality[1], consider the simplified scenario where the number of primitives $n$ is equal to the number of target concepts $y \in \mathbb{Y}$, and the following holds:

$$\forall y \in \mathbb{Y}, \exists! p \in \mathbb{C}, f \in \mathcal{F} \mid y = f(p) \tag{3}$$

In other words, each concept in the target text $\mathbb{Y}$ can be depicted uniquely as a transformation of a certain primitive, *i.e.*, there exists a one-to-one mapping between $\mathbb{C}$ and $\mathbb{Y}$ under $\mathcal{F}$. Now, consider adding $\eta$ additional primitives to the primitive set such that $\mathbb{C}$ now becomes $\{p_1, p_2, ..., p_n, p_{n+1}, ..., p_{n+\eta}\}$. However, the given premise states that generating $\mathbb{Y}$ was achievable using $\{p_1, p_2, ..., p_n\}$ only. Therefore the evolution of the additional primitives $\{p_{n+1}, ..., p_{n+\eta}\}$ is not constrained by the optimization objective, leaving open the possibility of them lying around the canvas as noisy strokes with no clear meaning. To guarantee sufficiency, *i.e.*, the condition in Eq. (3), we always overestimate the number of primitives required to visualize a certain text prompt, to ensure the synthesized sketch is complete with the required details. However, as argued formally, this overestimation could leave room for some noisy strokes crowding up the canvas. Primitive-level dropout ensures that this does not happen by randomly removing $\eta$ (proportional to the estimated noise rate) primitives in each iteration, thereby forcing all of the remaining primitives to contribute towards representing something meaningful. The empirical benefits of this PLD approach are discussed in Sec. 4.3 and can be seen in Fig. 5.

**Initializing Primitives with Diminished Opacity:** In traditional human sketching, artists often begin with a light outline or faint layout, serving as a foundation for the artwork. This initial phase sets the broader structure and composition. As the artwork advances, artists intensify strokes, especially focusing on crucial elements to make them prominent, ensuring each stroke adds value to the overall piece. Drawing a parallel to the digital realm, in CLIPDraw++, we have curated a similar methodology.

---

[1] In general, each concept would be depicted by multiple primitives, but the central claim of this formalization would still hold.

Here, the initialization of primitive shapes starts with a low opacity value, denoted as $\alpha$. This can be likened to the faint layout artists create. As the system begins its optimization process, based on relevance and significance, the opacity of certain primitives is incrementally increased. This mirrors the artist's method of iteratively intensifying strokes that are deemed crucial to the sketch's integrity. Formally, in each backward pass, we update the opacity value of a primitive $p$ as $\alpha_p \leftarrow \nabla_\mathcal{I}(\mathcal{L}_\text{total}); \quad \alpha_p > K$, where $\mathcal{L}_\text{total}$ is the optimization criterion of CLIPDraw++ from Eq. (4) and $K$ is an empirical constant. We retain $p$ in the canvas if the inequality is met, and drop it otherwise. In essence, CLIPDraw++ attempts to replicate the thoughtful and incremental approach artists employ, blending the nuances of human artistry with the precision of machine optimization. The advantage of initializing a canvas by strokes with diminished opacity is discussed in Sec. 3.2 and demonstrated in Fig. 7 of Supp.

### 3.4. Training Criteria

Our training criteria evaluate the alignment between input text prompts and the synthesized sketches, including their augmented versions. In order to measure the similarity, we employ the pre-trained text and image encoders from the CLIP model [24]. The ability of CLIP to encode information from both natural images (here sketches) and texts, eliminates the need for additional training. To measure the alignment of a given text and the synthesized sketch we use cosine similarity as $\text{sim}(x, y) = \frac{x \cdot y}{||x|| \cdot ||y||}$.

**Semantic Loss:** Our primary objective is to amplify the semantic similarity between the text prompts and the synthesized sketches. To measure the semantic congruence between a text prompt $P$ and rasterized version of the synthesized sketch $S$, we introduce the semantic loss function, $\mathcal{L}_\text{sem}$, designed to ensure semantic coherence and capture discrepancies between the two modalities.

$$\mathcal{L}_\text{sem} = -\sum_{i=0}^{M} \text{sim} \left( \mathcal{T}(P), \mathcal{I}(T_f(S)) \right)$$

where $\mathcal{T}_f$ indicates randomized affine transformations, and $M$ denotes the number of transformed variations produced through augmentations.

**Visual Loss:** While semantic loss focuses on high-level semantic cues, it may overlook important low-level spatial attributes such as pose and structure. Therefore, to complement the semantic loss, we introduce a criteria $\mathcal{L}_\text{vis}$ measuring the geometric congruence between the image generated by the diffusion UNet and the sketch synthesized by our model for the same input text, which is computed as:

$$\mathcal{L}_\text{vis} = -\sum_{i=0}^{M} \text{sim} \left( \mathcal{I}(I_0), \mathcal{I}(T_f(S)) \right)$$

where $I_0$ denotes the final image generated by frozen UNet.

**Total Loss:** The total loss, $\mathcal{L}_\text{total}$ is the summation of two loss functions (semantic loss $\mathcal{L}_\text{sem}$ and visual loss $\mathcal{L}_\text{vis}$) ex-

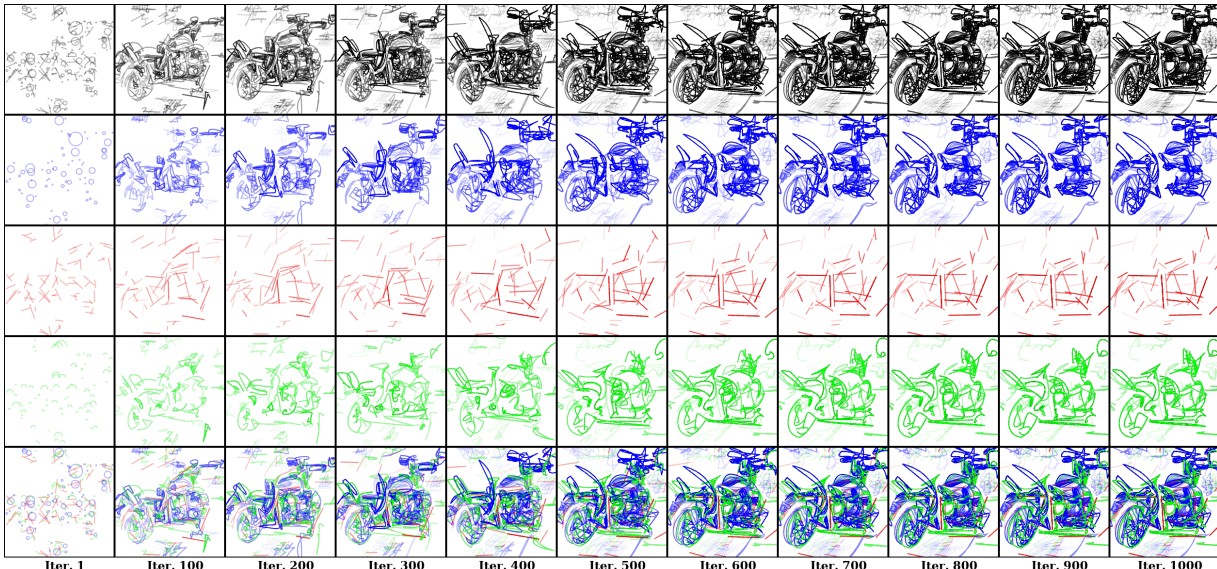

Figure 3. For the text prompt "A standing motorcycle ", our CLIPDraw++ tracks the evolution of each primitive shape during optimization: the first row shows a black-and-white synthesized sketch, the next three rows display the development of circles, straight lines, and semi-circles, and the final row combines these three rows' compositions. Here "motorcycle" is used to create the cross-attention maps.

plained above, each weighted by their respective coefficients, $\lambda_{\text{sem}}$ and $\lambda_{\text{vis}}$. These two loss functions balance our sketch synthesis process: semantic loss aligns vector sketches with textual prompts, while visual loss maintains low-level spatial features and perceptual coherence. This combination effectively captures the intricate relationship between semantic fidelity and geometric accuracy.

$$\mathcal{L}_{\text{total}} = \lambda_{\text{sem}}\mathcal{L}_{\text{sem}} + \lambda_{\text{vis}}\mathcal{L}_{\text{vis}} \qquad (4)$$

## 4. Experiments

In this section, we use CLIPDraw++ to synthesize sketches from linearly transformed primitives like circles, straight lines, and semi-circles. We also compare CLIPDraw++ with related methods and conduct ablations to evaluate its components. More results are in the supplementary.

### 4.1. Sketch Generation from Primitives

As shown in Fig. 3, we demonstrate that our CLIPDraw++ model offers the ability to synthesize sketches whose strokes are linearly transformed primitive shapes like circles (second row from the top), straight lines (third row) and half circles (fourth row). These individual strokes can be tracked through their evolution in successive iterations of the optimization process. Notably, the model intuitively represents different parts of the synthesized sketch with appropriate primitive shapes. For example, the chassis and handlebars of the synthesized *motorcycle* in Fig. 3 are rendered with straight lines, while the wheels are depicted using circles and semicircles. Our model skilfully captures the dynamics of shape or scene evolution, displaying varying levels of flexibility based on the degree of freedom, which is linked to the number of control points in a shape. In the *farm* ex-

ample (refer to Fig. 1 (b) of the supplementary), it assigns straight lines to simpler structures like a house's roof and walls, while more complex elements like grass and crops are made with semi-circles, providing more flexibility. Even more intricate structures, like trees, are rendered using circles, the most flexible shape, illustrating the model's skill in using various primitives for different levels of complexity. This strategic use of shapes enhances the model's ability to create detailed, nuanced sketches. Additional examples of sketch generation with primitive level tracking and overall sketch level tracking are respectively shown in Fig. 1 and Figs. 2-5 of the supplementary.

### 4.2. Comparison

Our CLIPDraw++ model is compared with five related methods: **CLIPDraw** [5], which optimizes Bézier curves for CLIP-guided sketches; **CLIPasso** [31], which simplifies sketches with Bézier curves; **CLIPascene** [30], which generates scene sketches from CLIP embeddings; **VectorFusion** [11], which uses a diffusion model for vector sketches; and **SVGDreamer** [33], which creates clearer sketches via a stroke-based diffusion approach. All models are tested with their original settings for fair comparison.

| Method / Metric | CS ↑ | PSNR ↑ | CLIP-T ↑ | BLIP ↑ | Conf. ↑ |
|---|---|---|---|---|---|
| CLIPDraw [5] | 0.2578 | 28.1740 | 0.3114 | 0.2611 | 0.49 |
| CLIPasso [31] | 0.2250 | 27.5000 | 0.2850 | 0.2783 | 0.45 |
| VectorFusion [11] | 0.2283 | 28.3277 | 0.2949 | 0.3894 | 0.44 |
| CLIPascene [30] | 0.2000 | 27.0231 | 0.2746 | 0.2551 | 0.42 |
| SVGDreamer [33] | 0.2688 | **28.7384** | 0.3132 | 0.4102 | **0.61** |
| **CLIPDraw++** (Ours) | **0.2763** | 28.6417 | **0.3365** | **0.4222** | 0.58 |

Table 1. Quantitative comparison with existing methods.

In Tab. 1, we present quantitative experiments to validate our approach using different evaluation metrics fol-

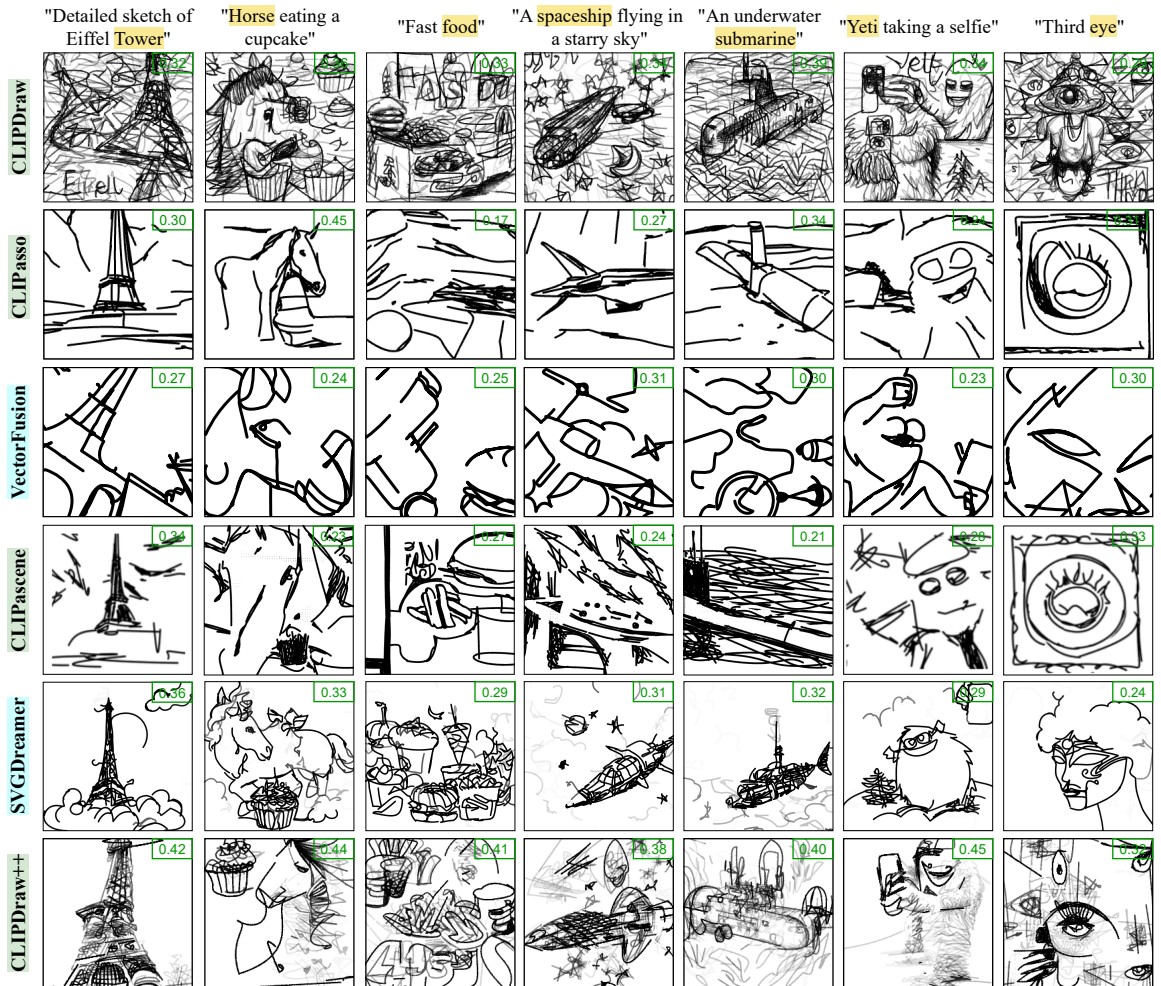

Figure 4. Qualitative comparison with quantitative measure between CLIP based approaches CLIPDraw , CLIPasso , CLIPascene , and diffusion based approaches VectorFusion , SVGDreamer , with our CLIPDraw++. The numbers in green at the top-right corner of each image indicate the CLIP-T score wrt the text prompt. The highlighted words are used to create cross-attention maps.

lowing existing literature [5, 33], including Cosine Simi-larity (CS) [10], Peak Signal-to-Noise Ratio (PSNR) [9], CLIP-T Score [24], BLIP score [13], and confusion score (Conf.) [32]. Our CLIPDraw++ has significantly outper-formed all other methods in CS, CLIP-T, and BLIP met-rics, and achieves the second best in PSNR and Conf, demonstrating its effectiveness. The higher scores in CS, CLIP-T, and BLIP indicate that CLIPDraw++ generates sketches closely aligned with the text prompts. Although SVGDreamer achieves slightly higher PSNR and confusion scores due to its stroke-based diffusion approach optimized directly in image space, our method is more efficient and achieves realistic sketches without extensive optimization. Furthermore, our high PSNR indicates less supersaturation, and the confusion score underscores the realism of our gen-erated sketches.

As demonstrated in Fig. 4, our CLIPDraw++ produces sketches that are noticeably cleaner and semantically closer (as indicated by CLIP-T score) than those from CLIPDraw,

likely due to our method's use of primitive-level dropout and the initialization of primitives at reduced opacity. Com-pared to CLIPasso, which simplifies sketches using Bézier curves but often loses fine details, CLIPDraw++ retains both clarity and detail without excessive smoothing. In con-trast to VectorFusion, which employs a diffusion model for vector sketches but struggles with abstract and less coher-ent representations, our method ensures more precise and semantically rich outputs. CLIPascene, while capable of generating scene-level sketches, often introduces clutter due to less effective control of primitive placement, whereas CLIPDraw++ maintains clarity through structured primitive initialization. Although SVGDreamer produces visually clearer sketches via a stroke-based diffusion approach, its optimization time is significantly longer, and it occasionally lacks semantic accuracy. In contrast, CLIPDraw++ consis-tently delivers clean sketches with accurate details and se-mantics. The reduced noise in our sketches is attributed to the minimized control points, learnable opacity for primi-

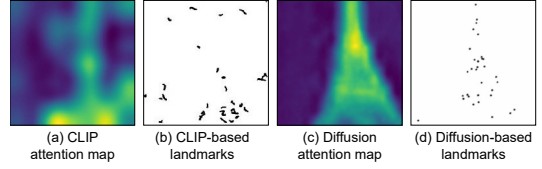

Figure 5. Effectiveness of primitive-level dropout (PLD) for the text prompt "Detailed sketch of Eiffel Tower ". The sketches in the top row are synthesized without PLD, while the ones in the bottom row are synthesized with PLD.

tives, and primitive-level dropout, thus reducing the need for manual intervention and parameter adjustments.

Figure 6. CLIP and diffusion attention maps and initializations.

### 4.3. Model Ablations

In this section, we present selected ablation studies of our model. Additional ablation results are in Sec. 3 of the Supp.
**CLIP-based vs Diffusion-based Initialization:** The non-convex nature of the optimization in CLIPDraw++ is sensitive to how primitives are initialized. We explore two methods: one using the CLIP attention map (Fig. 6 (a)) and the other based on the latent diffusion model's attention map (Fig. 6 (c)). Local maxima in these maps (Fig. 6 (b) and Fig. 6 (d)) identify key landmark points. Our findings show that the CLIP attention map lacks precision, spreading focus across irrelevant regions, while the diffusion model's attention offers more localized and detailed information. This precision is why we prefer the diffusion-based attention map for initializing sketches.

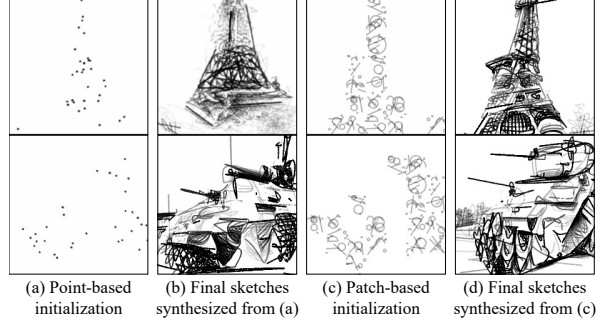

Figure 7. Comparison of results for 'Eiffel Tower' and 'Tank' for point-based and patch-based initialization.

**Patch-based Initialization:** In CLIPDraw++, we use a patch-based approach for stroke initialization, placing primitives within patches rather than directly at landmark points on the attention map. This method prevents the clutter and messiness typical of point-based initialization. As shown in Fig. 7, sketches created with patch-based initialization (Fig. 7 (d)) are clearer and more coherent compared to

those from point-based initialization (Fig. 7 (b)). Distributing primitives within a set range of attention local maxima (based on patch size) avoids excessive constraints and helps maintain clarity. This gives the optimizer a clearer view of the canvas, enabling more effective retention, evolution, or removal of primitives, resulting in cleaner sketches.

**Primitive-level Dropout:** Primitive-level dropout enhances the use of primitives by ensuring each encodes a specific concept, reducing noisy strokes that add no meaning. PLD also speeds up convergence by quickly identifying relevant strokes. As shown in Fig. 5, sketches with PLD (bottom row) are cleaner and more realistic, while those without PLD (top row) are noisier. To further assess PLD's effectiveness, we compared it with a strategy of gradually adding new strokes during optimization. In Fig. 8, ablation shows that PLD outperforms this approach in both visual quality and CLIP-T score. We suspect that abruptly adding new primitives disrupts the loss landscape's smoothness, making optimization harder and resulting in noisier sketches. More results are in Sec. 3.1 and Fig. 6 of the supplementary.

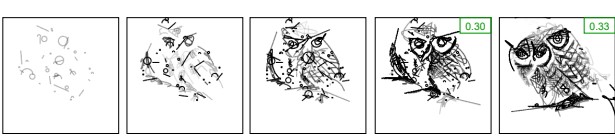

Figure 8. Comparison of efficacy of PLD with the strategy of sequentially adding extra primitives as required.

## 5. Conclusion

We introduced CLIPDraw++, a model for sketch synthesis through optimization using simple geometric primitives like straight lines, circles, and semicircles. Our model creates highly expressive sketches through simple linear transformations on these primitives, incorporating techniques such as strategic sketch canvas initialization for synthesizing clean sketches, and the introduction of primitive-level dropout for producing sketches with low noise, collectively enhance the model's efficiency and output quality. The extensive experiments and ablation studies underscore the model's superiority over existing methods, showcasing its ability to produce aesthetically appealing and semantically rich sketches. CLIPDraw++ excels in AI-driven art creation by merging advanced optimization with intuitive design principles.

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
