# CLIPDraw++: Text-to-Sketch Synthesis with Simple Primitives

## Paper ID 16

# Table of Contents

# 1. Implementation Details

In this section, we provide detailed information on how we have implemented our method. The implementation is carried out using `PyTorch` [3] and makes use of the differentiable rasterizer framework `diffvg` [2].

## 1.1. Initialization details

*"What is the procedure to initialize the first canvas?"* – to answer this question can be given in the following three steps: ❶ We first extract the crucial salient regions of the canvas by incorporating diffusion attentive attribution maps (DAAM) [5], leveraging the pre-trained Latent Diffusion model [4]. DAAM upscale and aggregate cross-attention word–pixel scores in the denoising subnetwork. For more details please refer to [5]. ❷ From the DAAM-generated final attention map, we sample $k$ positions. Here we have considered the value of $k$ to be 32. Thereafter, we follow patch-wise initialization as described within Sec. 3.2 in the main paper. Here, we divide the whole canvas into $32 \times 32$ patches and select only patches where any of $k$ points lie within, we do not initialize the primitives within these patches where there are no points belonging to them. A patch is denoted by $P_{i,j}$, where $i$ and $j$ represent the row and column, respectively. Here, $\forall (x, y) \in P_{i,j}$ lies within $(x_{i,j}^s, y_{i,j}^s)$, and $(x_{i,j}^e, y_{i,j}^e)$ where $s$ denotes the top-left/start point and $e$ denotes bottom-right/end point. A point, $(x, y)$ belongs to patch $P_{i',j'}$ where $(i', j') = ([x/32], [y/32])$. ❸ After choosing which patches to initialize the primitives, we randomly place exactly 3 primitives, (one from each type, straight line, circle, and semi-circle) within selected patches. Our primitives are based on the SVG (Scalable Vector Graphics) path but constrained to certain geometric shapes, described as follows:

- *Line:* A line can be represented by two control points, $l_1(x_1, y_1)$ and $l_2(x_2, y_2)$. To initialize a line, $L : (l_1, l_2)$ at the desired patch, $P_{i,j}$ we follow:

$$x_1 = \texttt{random.randint}(x_{i,j}^s, x_{i,j}^e)$$
$$y_1 = \texttt{random.randint}(y_{i,j}^s, y_{i,j}^e)$$
$$x_2 = \texttt{random.randint}(x_{i,j}^s, x_{i,j}^e)$$
$$y_2 = \texttt{random.randint}(y_{i,j}^s, y_{i,j}^e)$$

The length of line $L = \sqrt{(x_2 - x_1)^2 + (y_2 - y_1)^2}$.
- *Circle:* A circle, $C$ within a patch $P_{i,j}$ is represented by it's centre $\mathbf{c}$ $(x_c, y_c)$, and the radius $\mathbf{r}$ where

$$x_c = \texttt{random.randint}(x_{i,j}^s, x_{i,j}^e)$$
$$y_c = \texttt{random.randint}(y_{i,j}^s, y_{i,j}^e)$$
$$r = \texttt{random.randint}(1, r_{max}).$$

Here $r_{max}$ is the maximum size of the radius so that the circle should remain within the patch, $P_{i,j}$ which is defined in the below pseudo-code:

```
1  # randomly initialize (x_c, y_c)
2  max_radius_x = patch_size/2
3  max_radius_y = patch_size/2
4  if x_c < start_x + patch_size/2:
5      max_radius_x = x_c - start_x
6  else:
7      max_radius_x = end_x - x_c
8  if y_c < start_y + patch_size/2:
9      max_radius_y = y_c - start_y
```

```
10 else:
11     max_radius_y = end_y - y_c
12 max_radius = min(max_radius_x, max_radius_y)
```

where patch_size = 32; (x_c, y_c) is the center of the circle; and (start_x, start_y) = $\left(x_{i,j}^s, y_{i,j}^s\right)$ and (end_x, end_y) = $\left(x_{i,j}^e, y_{i,j}^e\right)$.

- **Semi-circle:** For the semi-circle, the mathematical explanation remains the same as the circle, the only change that occurs is in the SVG path and these can be approximated using a cubic Bezier curve.

The SVG path of these primitive types to incorporate `diffvg` library is given as:

```
1 import pydiffvg # diffvg library
2 # SVG line path
3 line_path = pydiffvg.from_svg_path(f'M {x1},{y1}
      L {x2},{y2}')
4 # SVG circle path
5 circle_path = pydiffvg.from_svg_path(f'M {x_c - r
      },{y_c} a {r},{r} 0 1,1 {r*2},0 a {r},{r} 0
      1,1 {-1*r*2},0')
6 # SVG semi-circle path
7 semi_circle_path = pydiffvg.from_svg_path('M {x_c
      - r},{y_c} a {r},{r} 0 1,1 {r*2},0')
```

This way we initialize the first sketch canvas using a set of simple geometric primitives before the optimization process begins.

### 1.2. Augmentation details

The primary aim of image augmentation is to preserve recognizability in the presence of various distortions. Following the implementation of CLIP-Draw [1], we utilize a series of transformation functions, namely `torch.transforms.RandomPerspective` and `torch.transforms.RandomResizedCrop`, on both the given image and generated sketch before passing them as inputs to the CLIP model for loss computation. In this context, the total number of augmentations, denoted as $N$, is set to 4. The incorporation of these augmentations serves to enhance the robustness of the optimization process against adversarial samples and contributes to an overall improvement in the quality of the generated sketch.

### 1.3. Optimization details

Our optimization loop includes the following three steps – 1) generating a sketch from vectorized primitives, 2) applying primitive-level dropout (PLD), and 3) computing loss function and back-propagate it to SVG parameters within `diffvg`. CLIPDraw++ optimization does not operate on width and color optimization, we keep them constant for the task of the sketch-synthesize. The optimization process typically takes around 2 min to run 1000 iterations on a single RTX 3090 GPU. Nevertheless, we attain a thorough grasp of the semantic and visual aspects of the provided textual prompt within 500 epochs. Yet, for the purpose of refining

it and enhancing noise removal, we continue the process for an additional 500 iterations as shown in Fig. 1. We employ the `Adam` optimizer along with the learning rate scheduler outlined in the following pseudo-code:

```
1 for t in range(num_iter):
2   ...
3   if t == int(num_iter * 0.5):
4       lr = 0.4
5   if t == int(num_iter * 0.75):
6       lr = 0.1
7   ...
```

Here, `num_iter` is the maximum number of optimization loops, we set it as 1000; `lr` is the learning rate, and we initially set it as 1.0.

> The words highlighted in yellow are used to generate the attention maps leveraging DAAM.

## 2. Sketch Generation from Primitives

As delineated in Sec. 3.1 and Sec. 4.1 of the main paper, our sketches are composed of linearly transformed primitives such as straight lines, circles, and semi-circles. These individual strokes can be tracked through their evolution in successive iterations of the optimization process. In this context, we present the visualization of sketch generation with the primitive level tracking in Fig. 1 and overall sketch level tracking in Figs. 2, 3, 4, and 5 after each 100 iterations, contextualized by diverse text prompts.

In the illustrative portrayal of *"Floating musical notes from a piano"* (refer to Fig. 1a), the composition employs straight lines for the base structure, and a combination of circles and semi-circles for floating musical notes. Analogously, in the depiction of *"Faucet"* (refer to Fig. 1d), straight lines define the wash basin, while the faucet's spout predominantly assumes a circular form, with its juncture to the basin evolving gracefully from semi-circular elements. Turning attention to the *"Supermarket"* scene (refer to Fig. 1e), straight lines form simple structures like shelves, contrasting with circles and semi-circles that compose more intricate elements such as displayed items.

Moreover, we present comprehensive results of sketch synthesis and their traceable versions for various input text prompts in Figs. 2 to 5.

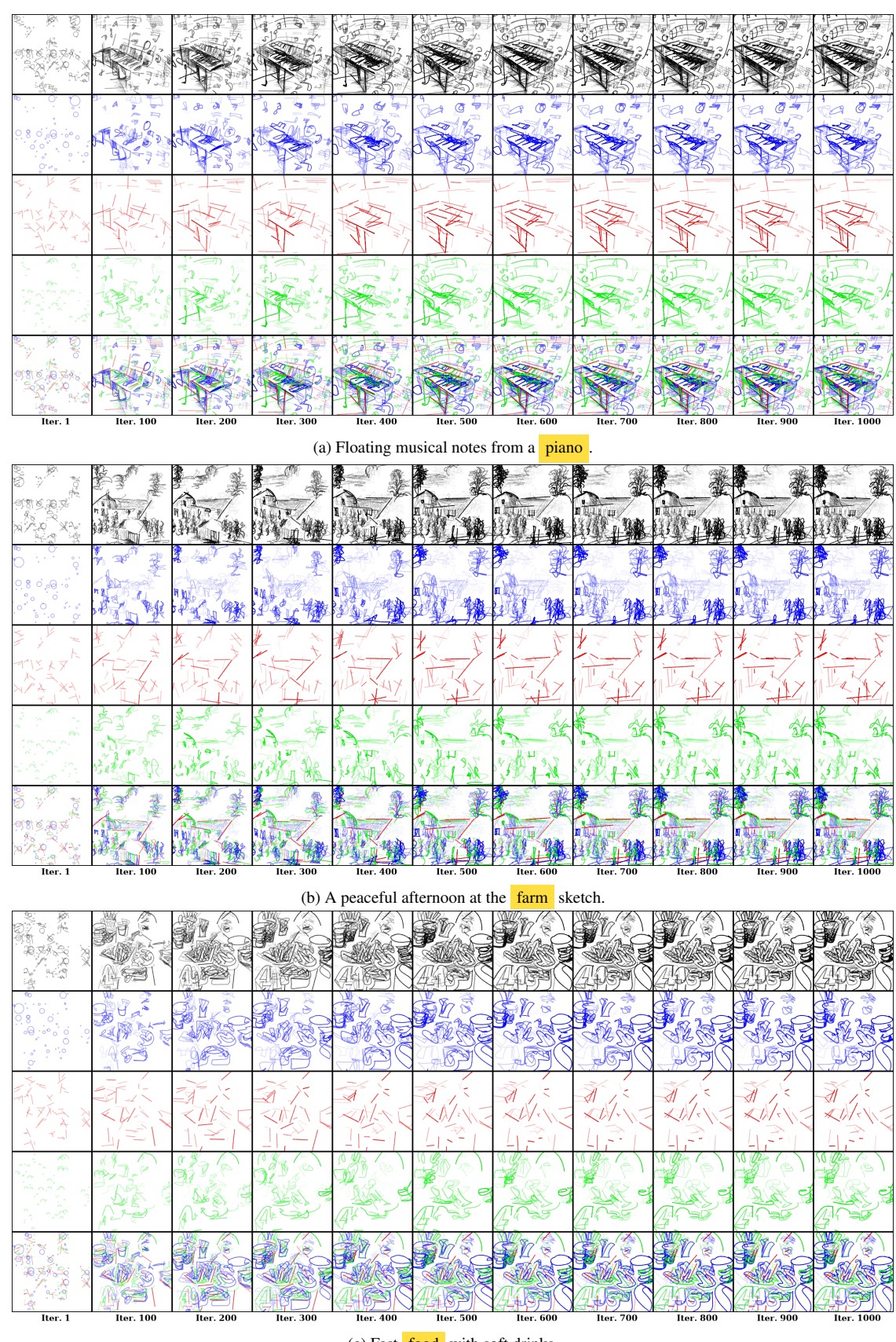

(a) Floating musical notes from a piano .

(b) A peaceful afternoon at the farm sketch.

(c) Fast food with soft drinks.

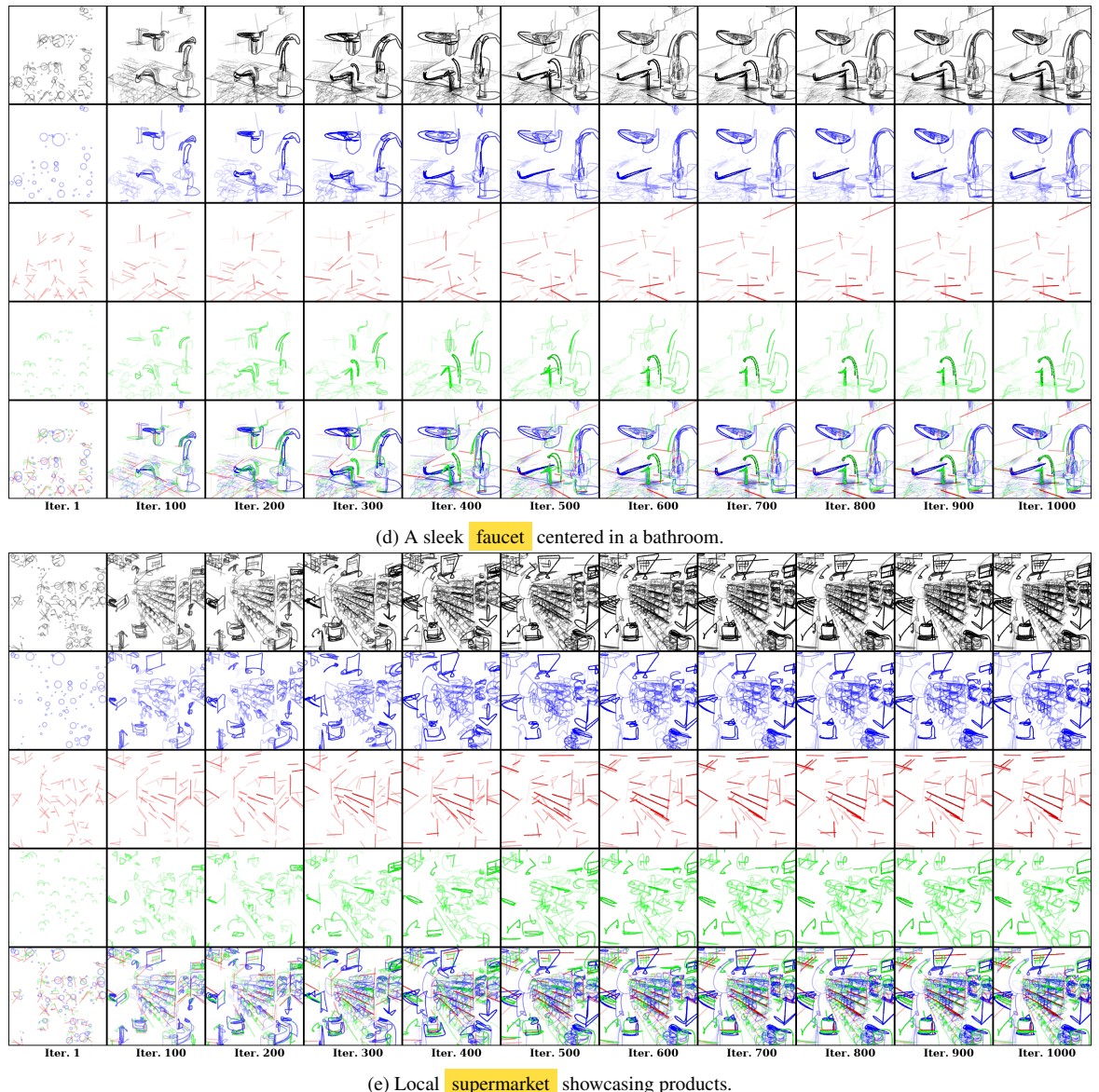

(d) A sleek faucet centered in a bathroom.

(e) Local supermarket showcasing products.

Figure 1. CLIPDraw++ illustrates the shape evolution of each primitive type in optimization: first row - black-and-white synthesized sketch, next three rows - circles, straight lines, and semi-circles, final row - combined compositions.

## 3. CLIPDraw++ Ablation Study

In this section, we showcase more examples to expound upon comprehensive ablation studies carried out on the different components of our CLIPDraw++ model.

### 3.1. Impact of Primitive-level Dropout

A thorough and intuitive mathematical explanation of Primitive-level Dropout (PLD) can be found in Sec. 3.3 of the main paper. Within this section, we demonstrate the robust effectiveness of PLD by presenting additional examples with varying dropout probabilities: 0 (without PLD), 0.05, and 0.1. Our findings indicate that a dropout probability of 0.05 yields sketches that are visually and semantically more complete and coherent compared to those without PLD and with a higher dropout probability, such as 0.1.

To illustrate, when using prompts such as *"A whimsical journey of a pig with a big nose and small legs"* and *"A ptarmigan's dance in the sky"*, we obtain a good finishing of pig's face and ptarmigan, respectively at a dropout probability of 0.05. As depicted in Fig. 6, sketches synthesized with a PLD of 0.05 generally exhibit cleanliness and realism, while those without PLD (top row) appear noisy and with a PLD of 1.0 occasionally exhibit incomplete struc-

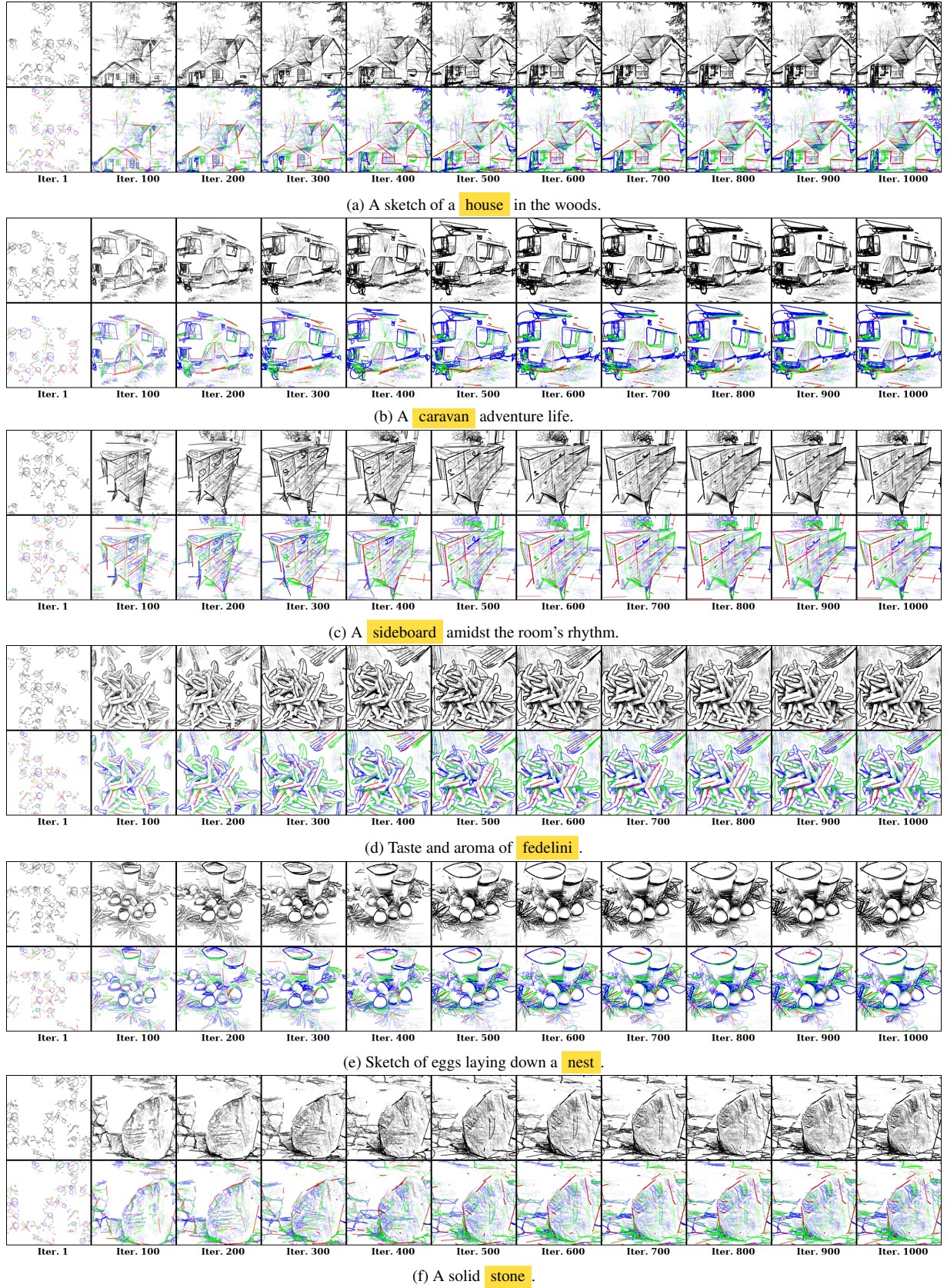

Figure 2. Visualizations of synthesized sketches and its traceable version w.r.t. varying optimization iterations.

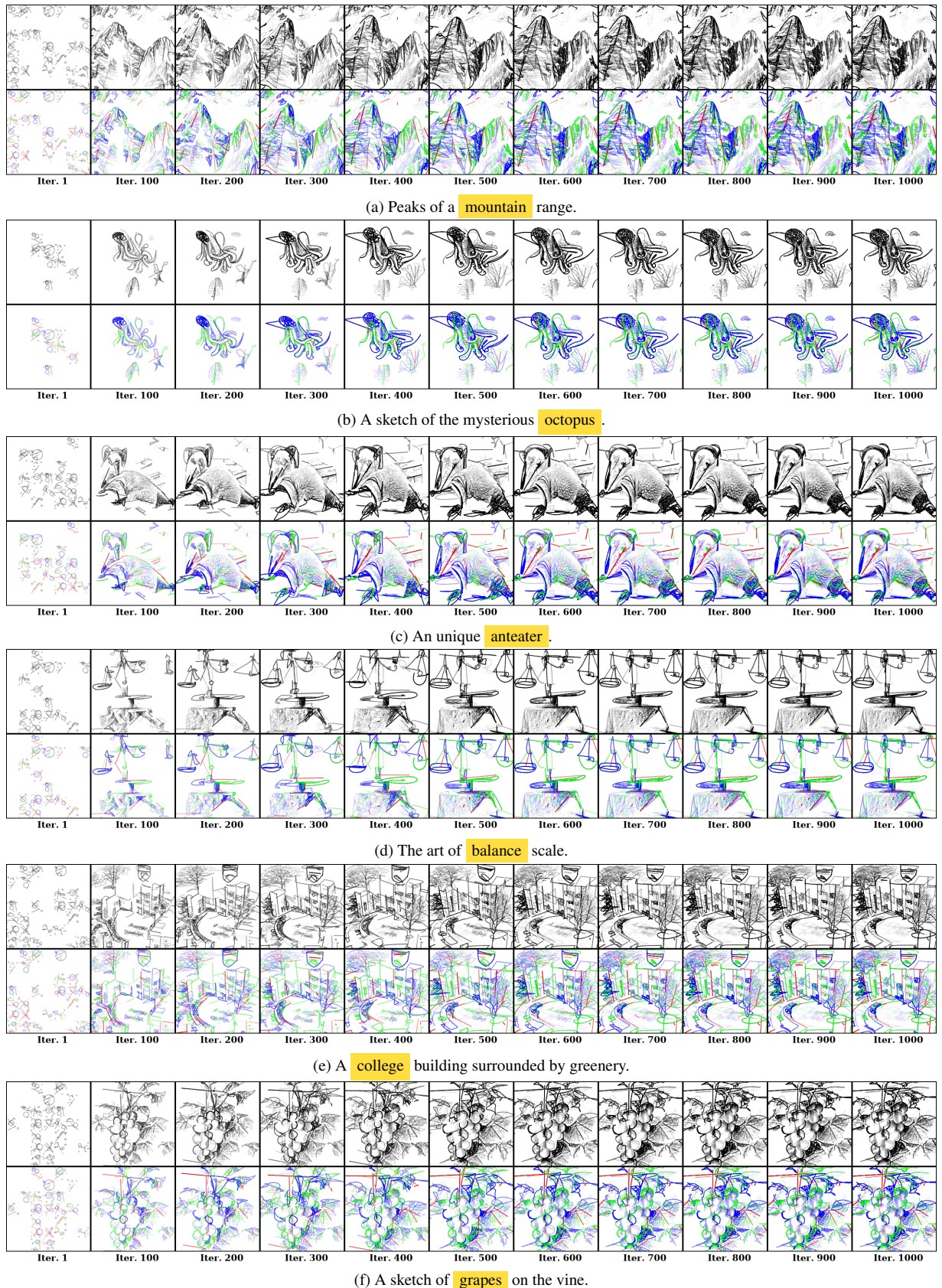

(a) Peaks of a mountain range.

(b) A sketch of the mysterious octopus.

(c) An unique anteater.

(d) The art of balance scale.

(e) A college building surrounded by greenery.

(f) A sketch of grapes on the vine.

Figure 3. Visualizations of synthesized sketches and its traceable version w.r.t. varying optimization iterations (continued to Fig. 2).

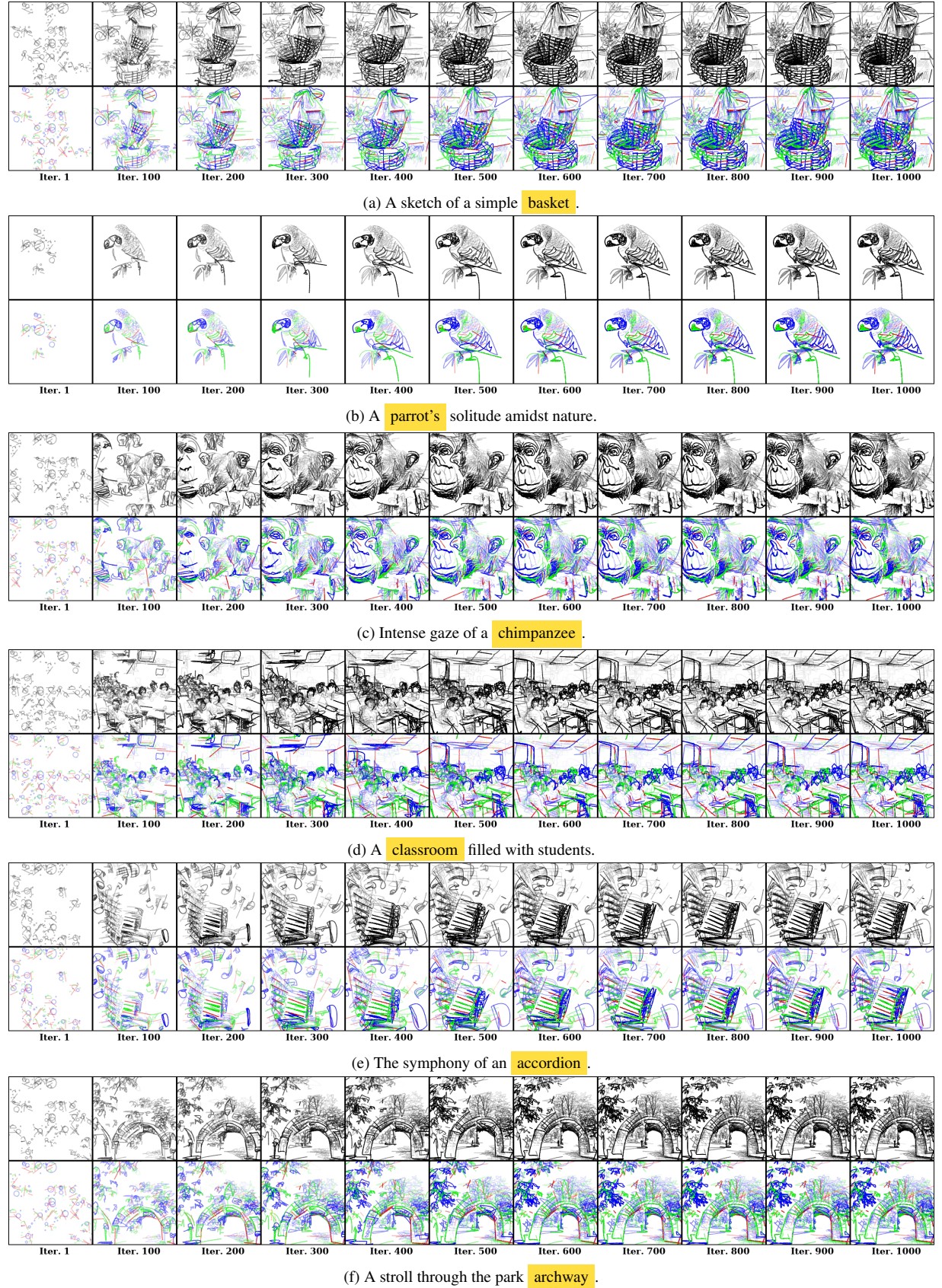

Figure 4. Visualizations of synthesized sketches and its traceable version w.r.t. varying optimization iterations (continued to Fig. 2).

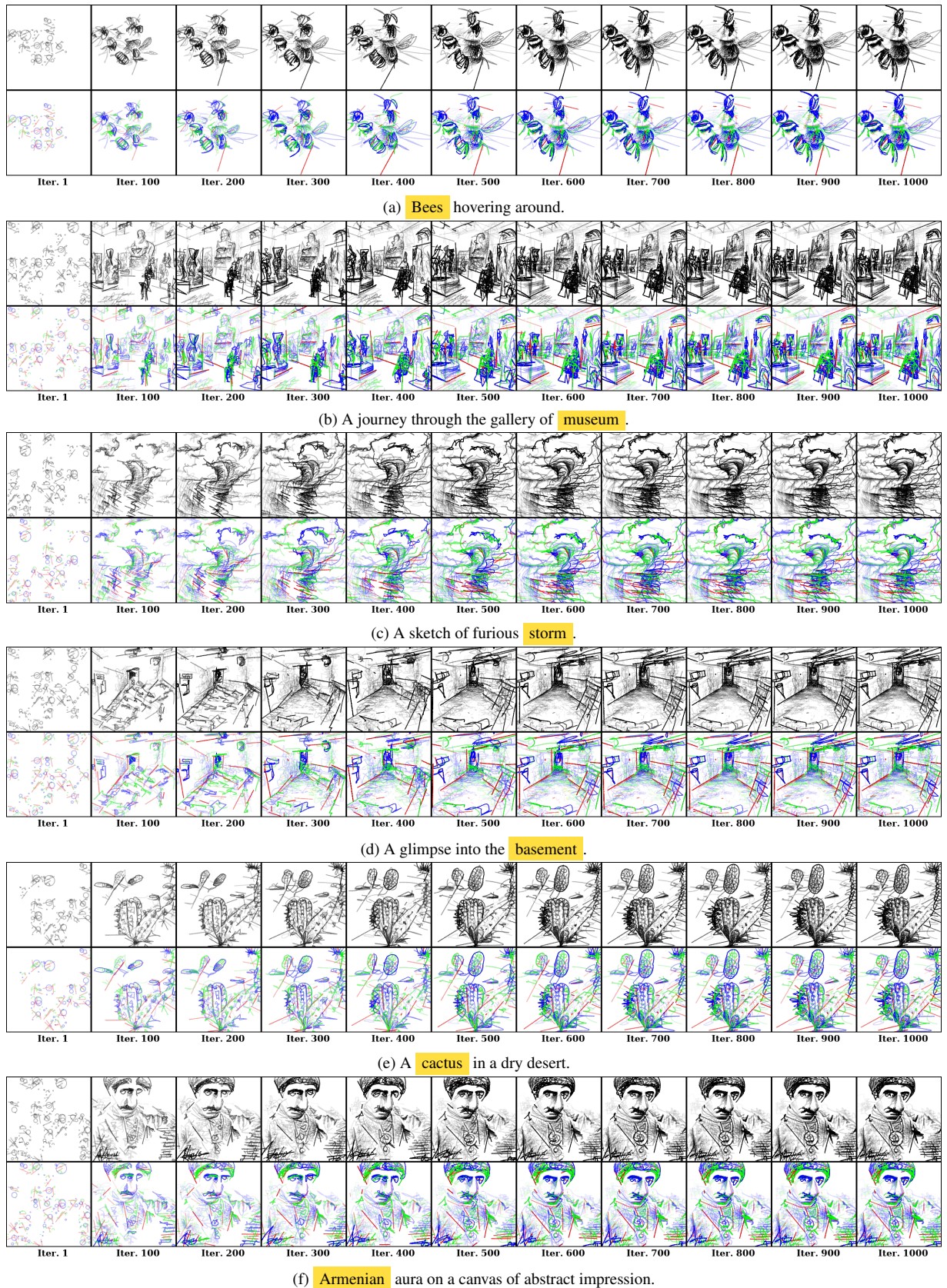

(a) Bees hovering around.

(b) A journey through the gallery of museum.

(c) A sketch of furious storm.

(d) A glimpse into the basement.

(e) A cactus in a dry desert.

(f) Armenian aura on a canvas of abstract impression.

Figure 5. Visualizations of synthesized sketches and its traceable version w.r.t. varying optimization iterations (continued to Fig. 2).

"Sketch of eggs laying down a nest"   "A whimsical journey of a pig with a big nose and small legs"   "A ptarmigan's dance in the sky"   "Floating musical notes from a piano"   "A cozy corner in the kitchen"   "A caravan adventure life"

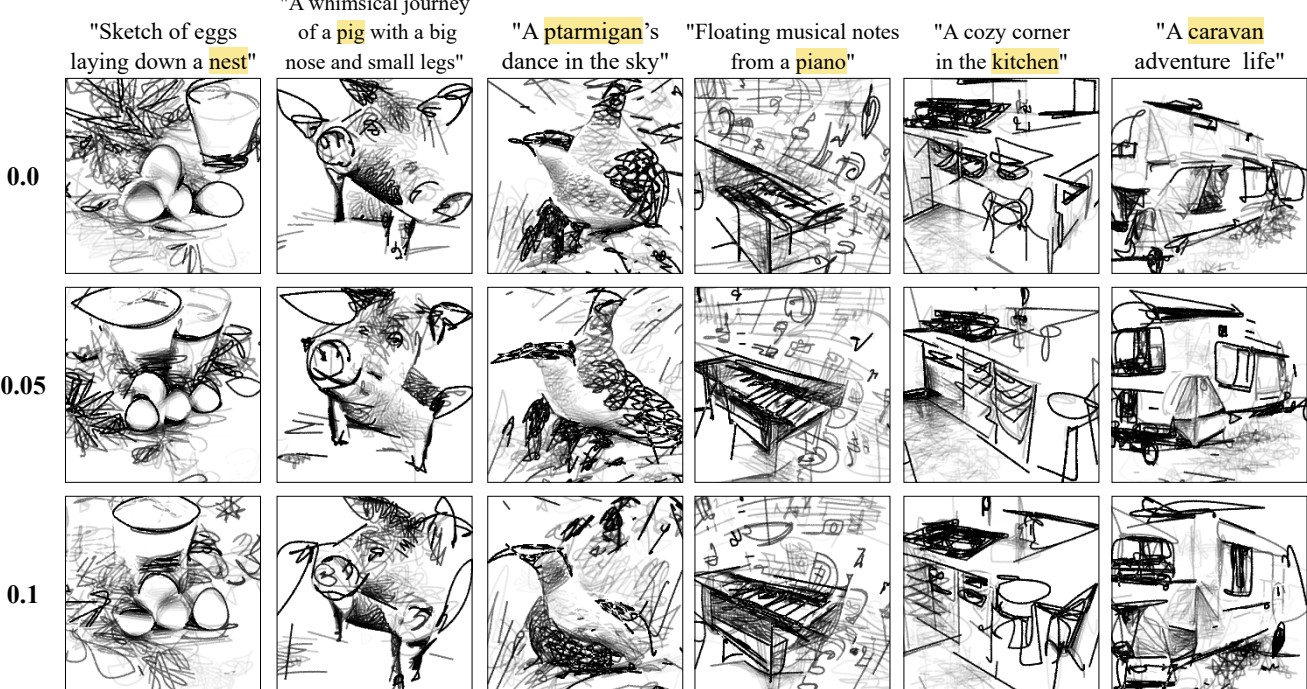

Figure 6. Effectiveness of primitive-level dropout (PLD) across various text prompts with top-row sketches generated without dropout, middle-row sketches with a 0.05 dropout probability, and bottom-row sketches with a 0.1 dropout probability.

ture in certain cases. This observation is also evident in *"Floating musical notes from a piano"* and *"A snug nook in the kitchen"* example where the structural completeness of the piano and the cozy kitchen corner are obtained with a PLD of 0.05. In the case of instances such as *"eggnog"* and *"caravan"* a PLD of 0.05 shows satisfactory outcomes but a PLD of 0.1 produces more promising results, highlighting the reliance of dropout probability on text prompts.

### 3.2. Sketching with Diminished Opacity

We start with highly transparent strokes (low opacity or a low $\alpha$ value) and gradually increase the opacity of only the essential strokes needed to convey the text prompt's semantics. In addition to primitive-level dropout, this approach further minimizes the presence of superfluous strokes in the synthesized sketches. As shown in Fig. 8, the final sketches which were initialized with lower $\alpha$ values are less noisy compared to the ones that are initialized with higher $\alpha$. By mimicking human drawing behaviour in this way, our approach demonstrates the potential to yield sketches that are more precise and finely crafted than those produced by conventional methods.

In Fig. 7, we present additional results demonstrating the impact of initiating sketching with diminished opacity, showcasing its influence on various text prompts. Our findings indicate that initiating primitives with $\alpha$ value 0.3 gives

the best semantically aligned and clearer results across all prompts, values lower than this threshold compromise semantic integrity, while values higher than this threshold lead to noisy distorted, and altered sketches.

### 3.3. Patch-based Initialization

In CLIPDraw++, we adopted a patch-based approach for initializing strokes, placing primitives in patches to cover areas within a certain range, instead of just at the exact landmark points identified on the attention map. This method of patch-based initialization is designed to prevent the cluttering and messiness often seen with point-based initialization. As demonstrated in Fig. 7 of main paper, sketches created through patch-based initialization stand out for their clarity and prominence. This approach allows for a more effective capture of the essential semantics of the input, resulting in cleaner, more coherent representations. In contrast, sketches originating from point-based initialization tend to be muddled and unclear. Allowing primitives to be evenly distributed up to a certain distance of the attention local maxima (determined by the patch dimensions) prevents the model from being overly constrained, and also maintains clarity in each of the local regions at initialization. This helps the optimizer have a much clearer view of the canvas at the start, which, in turn, lets it retain, evolve, or drop primitives based on semantic requirements with greater ease

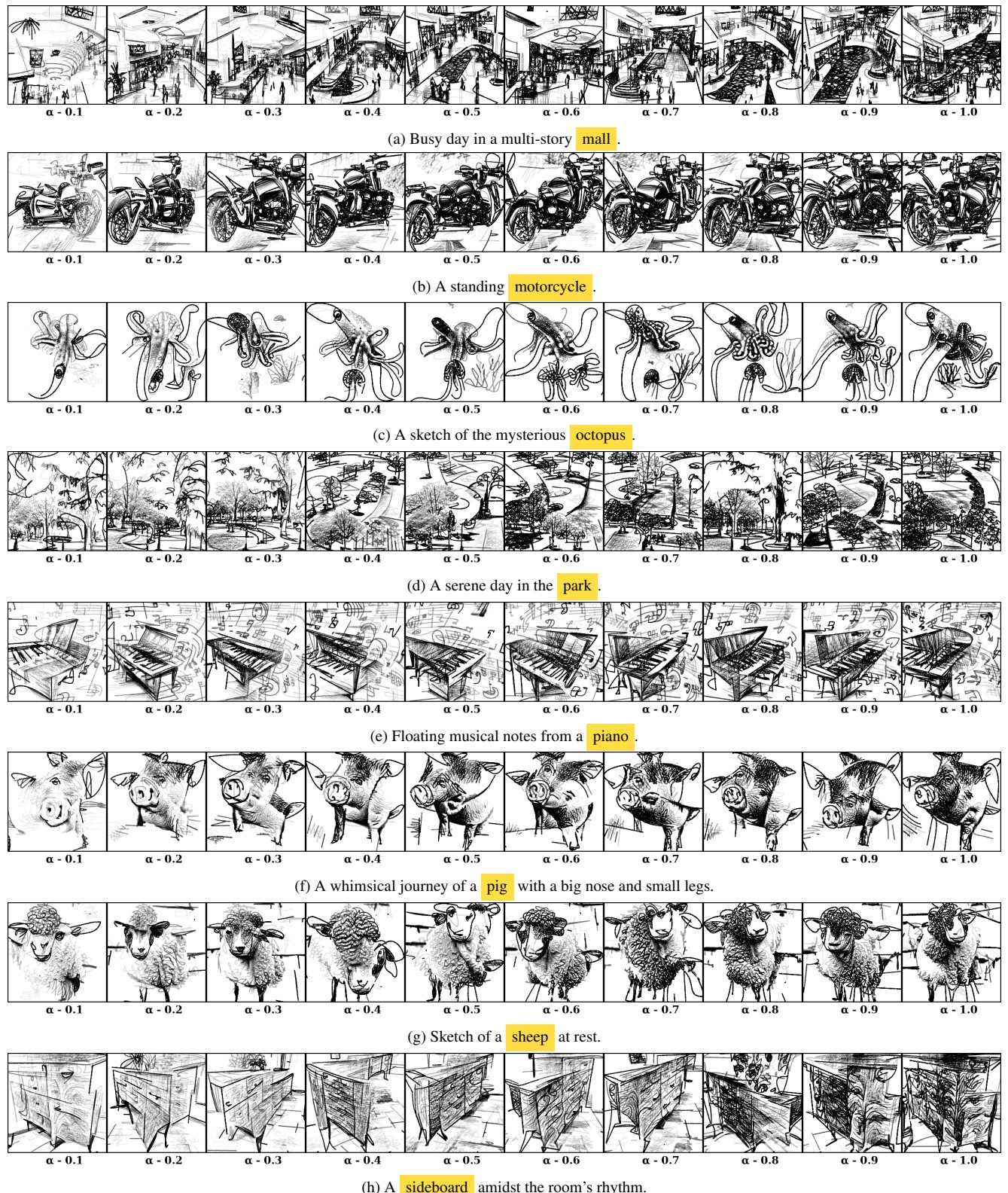

Figure 7. Effectiveness of initializing primitives with diminished opacity. Initiating primitives with lower $\alpha$ yields cleaner final sketches compared to higher $\alpha$. We report optimal $\alpha$ value as 0.3.

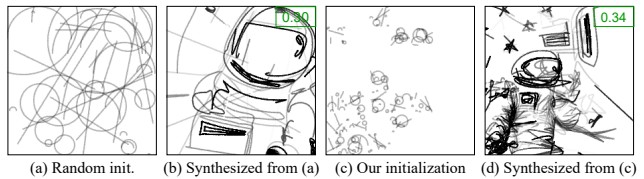

Figure 8. Effectiveness of initializing primitives with diminished opacity, indicated by lower $\alpha$ values, is notable. Initiating primitives with lower $\alpha$ in the prompt "A missile ready for launch" yields cleaner final sketches compared to higher $\alpha$.

– all of which eventually contribute to much clearer output sketches.

In Fig. 9, we demonstrate the superiority of our patch-based initialization compared to random initialization, with the former adhering to our strategy and the latter to the CLIPDraw approach for initially setting up canvases. Both qualitatively and quantitatively, our patch-based method outperforms the random approach.

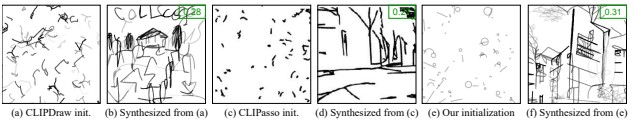

Figure 9. Random vs. patch-based (Ours) strokes initialization.

In Fig. 10, we compare our patch-based initialization against the random initialization of CLIPDraw [1] and CLIPasso [6], using an equal number of Bézier curves/primitives for canvas initialization. Our patch-based technique surpasses the approaches of CLIPDraw and CLIPasso in both quality and performance, even with the same number of strokes.

Figure 10. our patch-based initialization in comparison with CLIPDraw and CLIPasso based initialization with same number of strokes (primitives).

### 3.4. Analysis of Patch Size

The effectiveness of patch-based initialization in comparison to point-based initialization is detailed in Sec. 3.2 and illustrated in Fig. 7 of the main paper. Here, we explore the influence of different patch sizes during primitive initialization. For this purpose, we divide a $224 \times 224$ canvas into smaller patches, with dimensions of $32 \times 32$ and $56 \times 56$. It is evident from the Fig. 11 that, for the majority of the prompts, the $32 \times 32$ patch exhibits more visual details and aligned textual semantics compared to the $56 \times 56$ patch

size (the highest difference between them is being observed for prompts *"A sketch of a cauliflower"* and *"A caravan adventure life"*), with the exception of the *"The wise owl's gaze"* example where the $56 \times 56$ patch size displays more structural details of the owl.

### 3.5. Effect of primitives count within a patch

CLIPDraw++ initializes the primitives within selected patches. In context, it becomes very crucial to study how many primitives from each type (straight line, circle, and semi-circle) we need to initialize within the selected patches. In Fig. 12, we delve into the impact of varying the number of primitives in each category on the synthesis of sketches using diverse text prompts. Here, our findings unfold as follows – (1) When the primitive count is low for each type as seen in the first and second rows of Fig. 12, CLIPDraw++ tends to generate abstract representations to the provided text prompts. (2) Increasing the primitive counts to around 3 or 4 for each type (third and fourth rows in Fig. 12) results in more detailed drawings that incorporate additional features. (3) However, an increase in primitive counts beyond a certain threshold, as depicted in the fifth row of Fig. 12, does not necessarily lead to improved sketch synthesis. Such increments introduce intricacies into the optimization process, making it more complex, time-consuming, and resource-intensive. Given the fact that for simpler prompts many of these primitives are superfluous for sketching, which may yield sub-optimal outcomes, as well.

Keeping the above-mentioned findings in mind, we consider deploying 3 primitives from each type within the designated patches, which aims to constrain the optimization process to a suitable image space while utilizing the minimum number of primitives or strokes necessary.

### 3.6. Loss Ablation and Analysis

Our total loss function, $\mathcal{L}_{\text{total}}$ is composed of two loss functions, semantic loss: $\mathcal{L}_{\text{sem}}$ and visual loss: $\mathcal{L}_{\text{vis}}$, weighted by their respective coefficients or importance, $\lambda_{\text{sem}}$ and $\lambda_{\text{vis}}$.

$$\mathcal{L}_{\text{total}} = \lambda_{\text{sem}}\mathcal{L}_{\text{sem}} + \lambda_{\text{vis}}\mathcal{L}_{\text{vis}}$$

A detailed explanation of the aforementioned loss function and its components are given in Sec. 3.4 of the primary

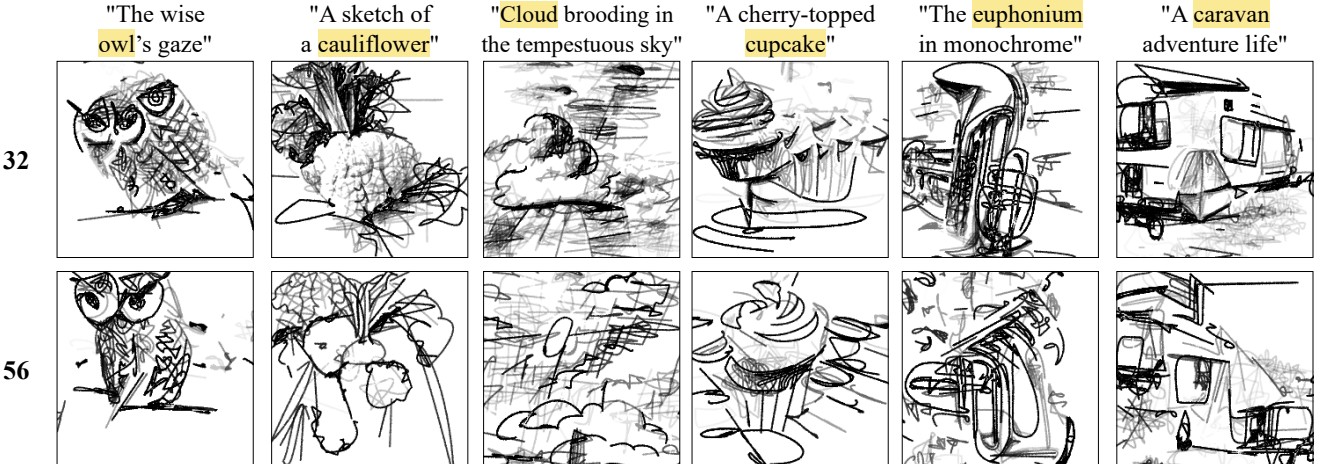

Figure 11. Comparative analysis of patch sizes ($32 \times 32$ vs. $56 \times 56$) in primitive initialization. The $32 \times 32$ patch consistently enhances visual details and textual coherence across most of the prompts.

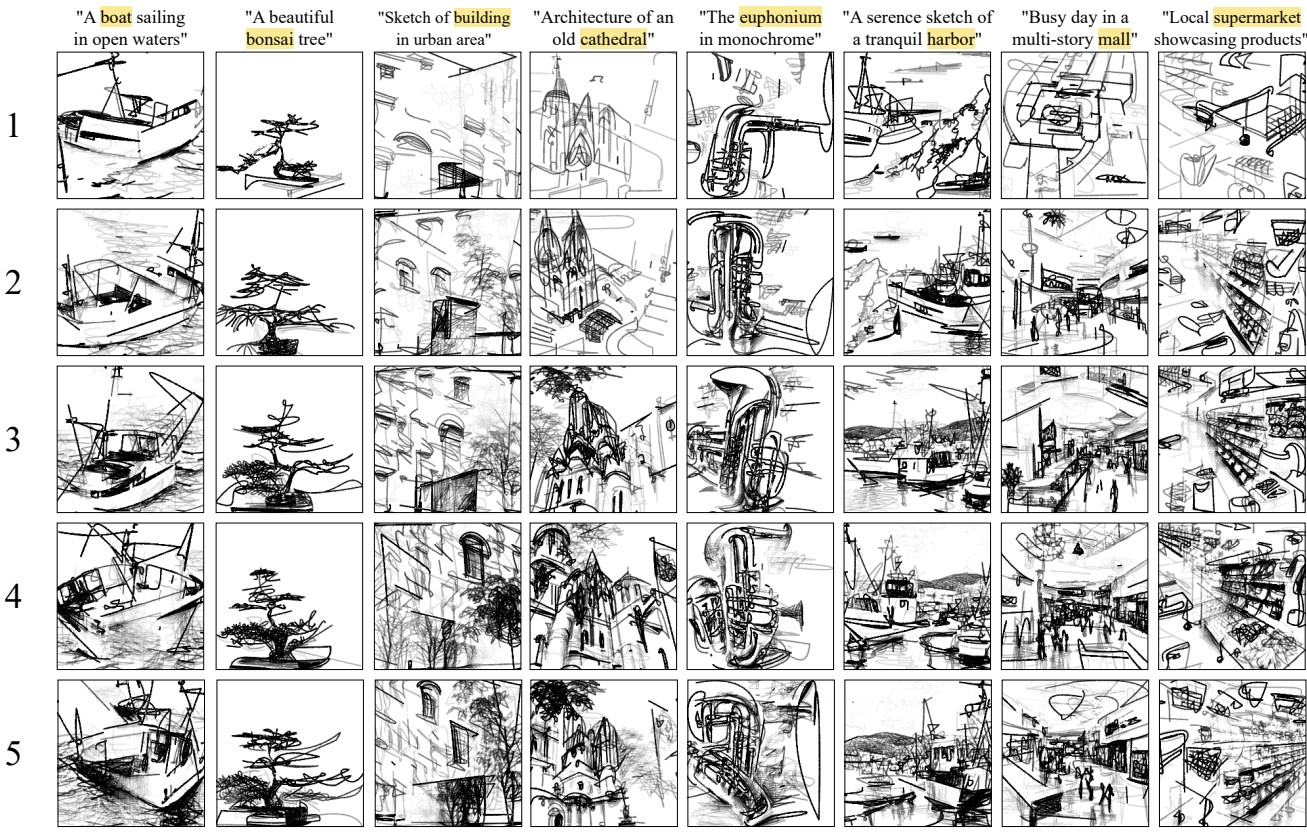

Figure 12. Exploring the impact of varying primitive counts for each type within selected patches on sketch outcomes with diverse text prompts in CLIPDraw++. (1) Few primitives yield abstract sketches (rows 1-2). (2) Optimal detail emerges with 3-4 primitives (rows 3-4). (3) Excessive counts (row 5) complicate optimization, hindering synthesis. The study recommends deploying 3 primitives per type for efficient synthesis, balancing complexity and resource utilization.

manuscript. Within Fig. 13, we present two sketches corresponding to the prompts *"A portrait of an actor"* and *"Apples hanging in full bloom"* while varying weight factors $\lambda_{\text{sem}}$ and $\lambda_{\text{vis}}$ individually. For the "actor" example in

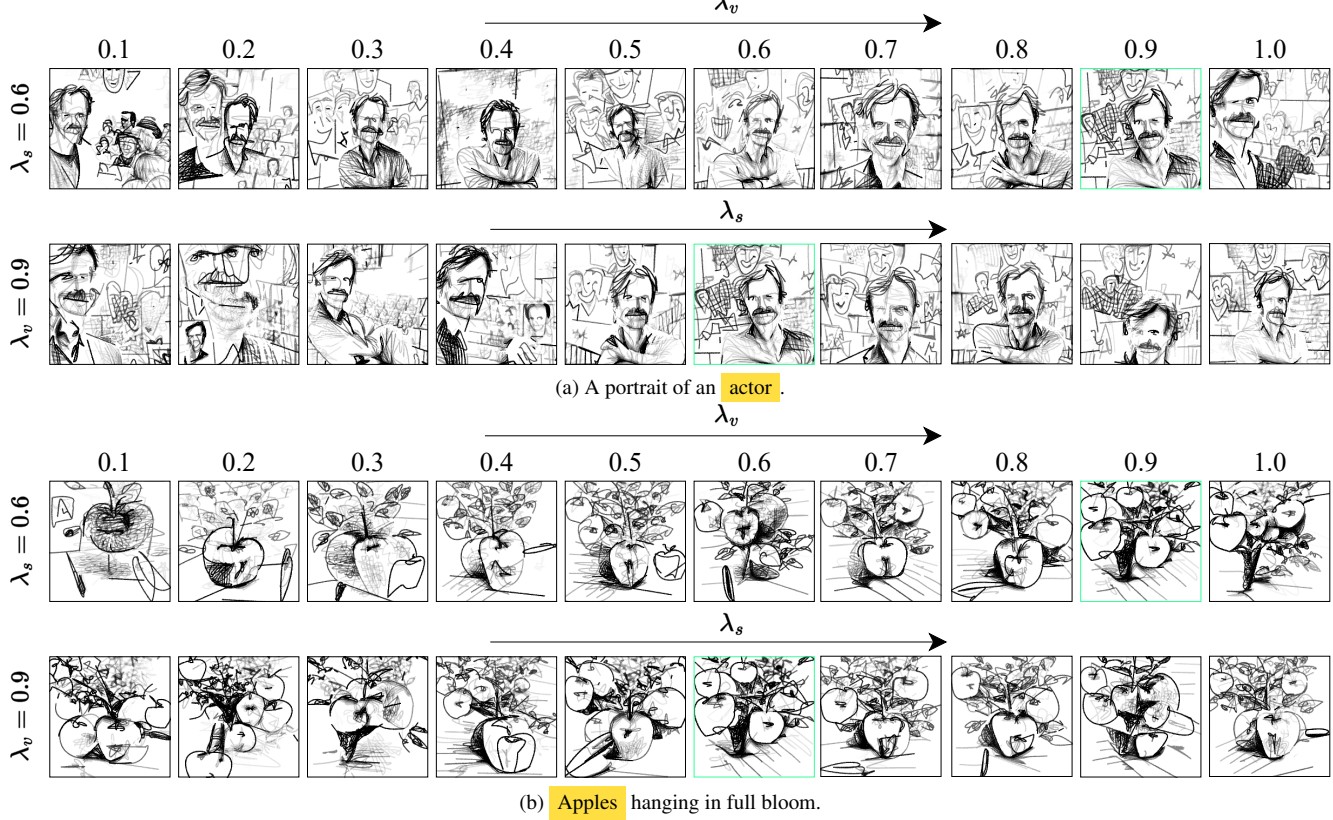

Figure 13. Ablation study on the impact of semantic and visual loss weights, $\lambda_{\text{sem}}$ and $\lambda_{\text{vis}}$ in the total loss function. Sketches reveal optimal values of 0.6 and 0.9, respectively, for $\lambda_{\text{sem}}$ and $\lambda_{\text{vis}}$.

Fig. 13a, we fix $\lambda_{\text{sem}}$ as 0.6 and vary $\lambda_{\text{vis}}$ values while in the second row, we fix $\lambda_{\text{vis}}$ and vary $\lambda_{\text{sem}}$ to show the impact of each components. Our analysis establishes the optimal values for $\lambda_{\text{sem}}$ and $\lambda_{\text{vis}}$ at 0.6 and 0.9, respectively. Utilizing these prescribed values yields a meticulous representation of the upper body structure for the actor (in Fig. 13a) and an accurately contoured depiction of the apple (in Fig. 13b) example.

### 3.7. Variations

Being a generative sketch synthesis model, our CLIP-Draw++ generates diverse sketches contextualized by the same text prompt. These enhanced variations are attributed to the utilization of robust latent diffusion models during the initialization stage. Fig. 14 provides evidence for the afore-mentioned claims, displaying four different output sketches generated in response to the same input description.

### 3.8. Advantage of Primitives over Bézier Curves

In Fig. 16, we compare the effectiveness of using primitive shapes like straight lines, circles, and semi-circles, with Bézier curves, where we use Bézier curves in combination with our remaining novel methodological components, and observe that the Bézier curves still result in noisier sketches (lower CLIP-T scores) compared to our primitives. We conjecture that since Bézier curves are more general objects, they suffer from the lack of any geometric inductive bias (which, in this case, is that, sketches can just as well be expressed via simpler primitives like straight lines, circles, and semi-circles). Therefore, they require wastefully more strokes, making the synthesis task *harder to optimize*, and one that results in significantly more *noisy* sketches.

## 4. Limitations

In this section, we shed some light on the proposed method's limitations, showcasing failure cases and exploring possible reasons for such occurrences as follows:

- **Interpretation of text prompts**: LDM (used to compute DAAM maps) and CLIP are not trained on similar objectives, which could lead to discrepancies in the way they process and interpret the input signals. In Fig. 15, the sketch of *"An angry woman staring at coworker"* might not depict the emotion of a woman as being angry or the context of her staring at a coworker.
  This issue can be somehow tackled by generating a vari-

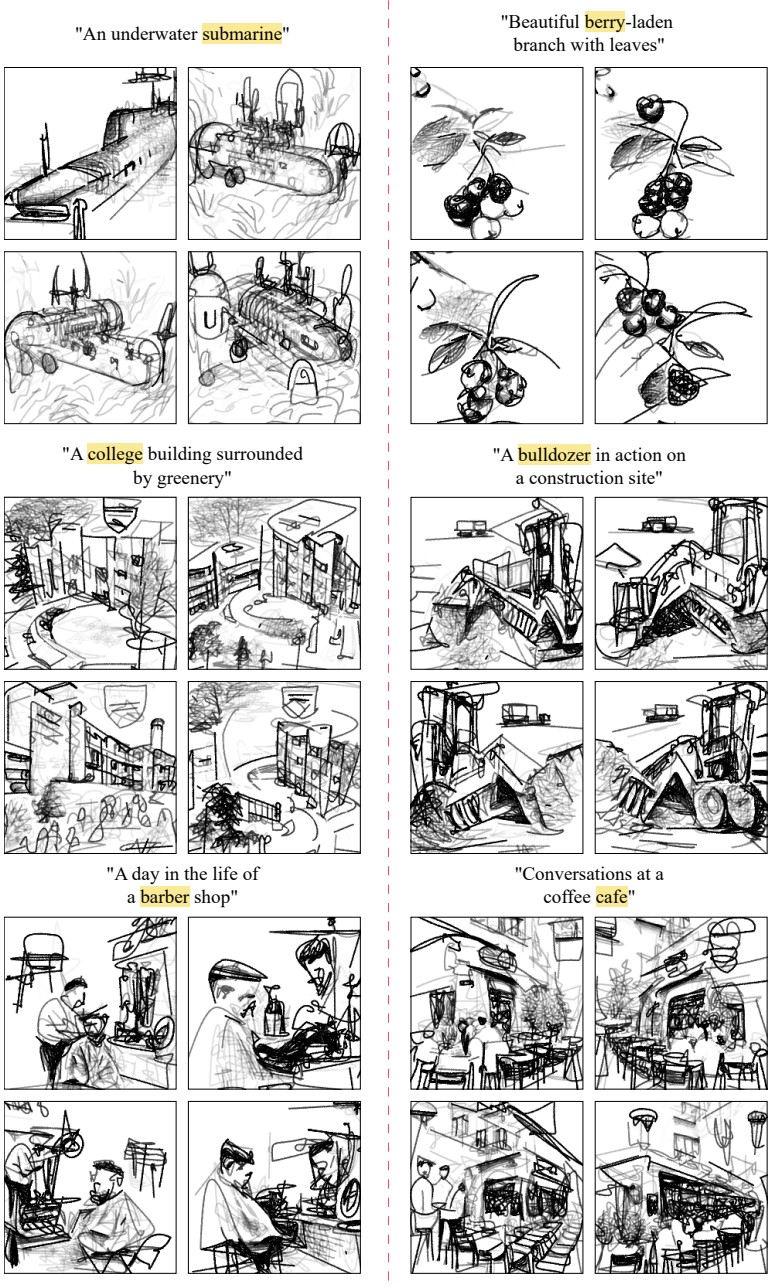

Figure 14. Our proposed CLIPDraw++ method offers the generation of diverse sketches while adhering to their corresponding textual semantics. In each example, given the same text prompt, four different sketches are synthesized using random seeds.

ety of cross-attention maps through the alteration of the optimizer's *SEED*.

- **Lack of details**: The sketches might lack the necessary detail to fully convey all the contexts within a prompt. For instance, the *"Sketch of a tiny mechanical clock"* (Fig. 15) might not clearly show the mechanical aspects of the clock. Similarly for *"Poster of an anime show describing various characters"* does not describe all the characters precisely.

The complexity of the sketches generated can vary based on the primitive count. Increasing the number of primitives can generate more complex visual representations and textures for the input text prompt.

"An angry woman staring at coworkers"

"Sketch of a tiny mechanical clock"

"Poster of an anime show describing various characters"

"A bagpiper playing in a wedding"

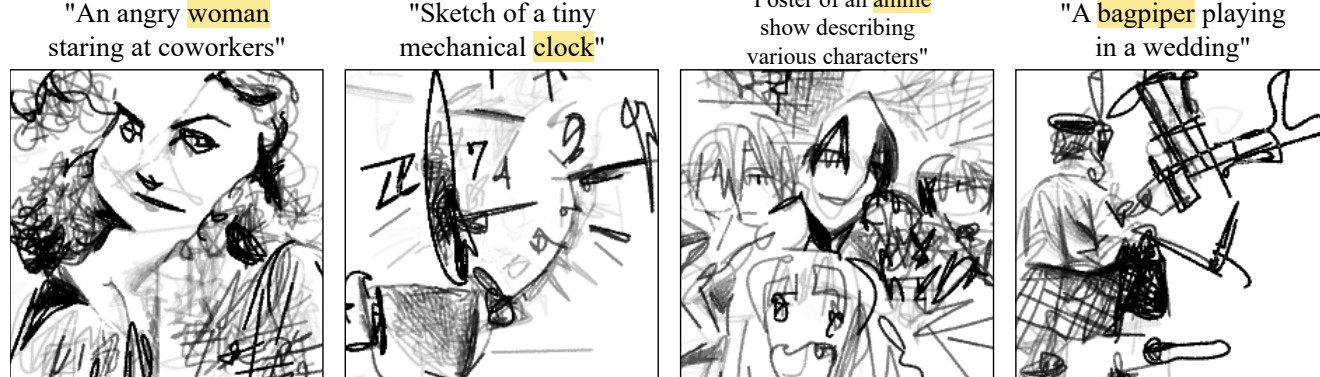

Figure 15. Some instances where the CLIPDraw++ model failed to synthesize sensible results.

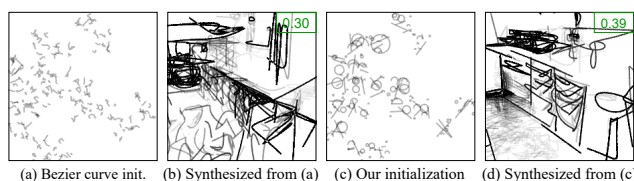

(a) Bezier curve init. (b) Synthesized from (a) (c) Our initialization (d) Synthesized from (c)

Figure 16. Advantage of using primitives over Bézier curves.