# OpenReview forum: "CLIPDraw++: Text-to-Sketch Synthesis with Simple Primitives"
_thecvf.com/CVPR/2025/Workshop/CVEU — CVPR 2025_

### Official Review · Reviewer_tr8f · 2025-03-21
**This paper offers a significant and original contribution to the field of vision-language generation by marrying text-to-sketch synthesis with explainability. It addresses an important problem – making AI-generated art interpretable – and does so with creativity and technical proficiency. The approach is novel, the methodology is solid and well-validated, and the results clearly demonstrate advantages over prior work. The writing is generally clear and the authors have been thorough in evaluation. The few weaknesses identified are not fundamental flaws but rather opportunities for further improvement. They do not overshadow the paper’s strengths. In line with the CVPR reviewing guidelines, I note that the paper has strong novelty and potential impact.**

**Rating:** 5
**Confidence:** 4

**Review:**

Review for CLIPDraw++

Summary:
The authors formulate the sketch generation as an optimization problem guided by CLIP embeddings, similar in spirit to CLIPDraw, but extend it with multiple new components. In this approach they simplify the computational complexity by using primitive shapes like circle, semi-circle and lines instead of the traditional Bézier curves. The method also models the distribution of these primitives using attention heatmaps from pre-trained diffusion models to further simplify the problem and uses a dropout on these primitives to boost the significance of each primitive. They also leverage an additional visual loss to get signals from an actual image strengthening the structural integrity of the sketch. The paper compares the resulting sketches with the existing approaches and shows considerable improvement in the sketches in terms of quality and clarity.

Strength:

Novelty and Originality:
This work proposes a novel approach to text-to-sketch synthesis by using simple geometric primitives (lines, circles, semicircles) instead of free-form curves, marking a clear departure from prior methods. This is a fresh idea that yields an inherently simpler and more interpretable solution.  The integration of diffusion model cross-attention for initialization of the primitives is another original component – to the best of my knowledge, no prior text-to-sketch method combined CLIP-based optimization with diffusion-based attention maps. The introduction of primitive based dropout is a neat addition that forces each primitive to carry a meaning on its own which ensures that each stroke is purposeful.

Technical Depth and Rigor:
The technical approach in this paper is sound and thorough. The paper formulates the problem in clear terms and provides intuition behind each element mentioned above. This intuition is backed by an extensive ablation work to provide evidence for each design choice like:
- the impact of Primitive Level Dropout on the quality of the output
- the efficacy of simple primitives against the Beizer curves
- the patch based approach vs landmark based for primitive initialization
- the impact of opacity scheduling of primitives for getting the most clean sketches
Each of these components is validated either qualitatively or quantitatively. Crucially, the paper’s claims are backed by evidence. For example, the claim that the method produces cleaner, more detailed sketches than prior methods is supported by both visual comparisons and an explanation that fewer control points and PLD reduce noise.

Comparison with Prior Work:
The paper acknowledges the most relevant prior works and makes fair comparisons to them. The qualitative examples provided support their claims. They show a clear improvement in terms of clarity and structural integrity and lack of noise. It’s impressive given the small count of primitives being used (3) for each patch. The authors provide clear reasons for these improvements as well.

Reproducibility:
The paper provides many implementation details in both the main text and the supplementary material.  The optimization process is described step-by-step, including initialization strategies (using diffusion cross-attention to place primitives in likely relevant regions) and training tricks (data augmentation, dropout). In the Appendix, there are sections where the authors have documented the needed information (such as learning rates, number of iterations, CLIP model version, etc.). This level of detail should allow an informed reader to reimplement the method without much guesswork. The intent to release the code upon final approval shows a commitment to open-source the work.

Minor areas of improvement

Runtime and efficiency:
One major claim is the efficiency of the primitive based sketch synthesis as compared to prior works using Bézier curves. While intuitively it makes sense, it would have been great if this was presented in a quantitative analysis of the memory and compute usage against the prior methods. CLIP-based optimization can be slow (CLIPDraw typically required hundreds of iterations). Suggestion: Provide some information on the runtime or convergence speed. For instance, “our method converges in X iterations (Y minutes on a GPU), which is 2× faster than CLIPDraw” (if true). If the method is actually slower due to extra overhead (diffusion, multiple losses), acknowledging that and suggesting future optimization (like a possibility of a learned initializer or using better optimizers) would be honest. This doesn’t detract from the contribution but sets realistic expectations for practical use.

Scalability and Complex Prompts:
The method was showcased on a variety of prompts, but it’s not clear how it scales with very complex scenes or long paragraphs of description. The reliance on a fixed number of primitives per type (e.g., 3 of each) could become a limitation if the prompt describes many distinct objects or requires fine detail. Suggestion: It would be useful to discuss the method’s limitations in such cases. Can it handle, say, a paragraph describing a detailed cityscape? If not, clarifying the scope (perhaps it’s aimed at single-scene or single-object sketches) is important. The authors might consider adding results or discussion on failure cases when the description is extremely complex, and perhaps suggest how the framework could be extended (e.g., by increasing primitive count or introducing new primitive types like rectangles or curves) for more complex inputs.

Minor Clarity Issues:
One minor clarity issue is that the paper introduces an innovative use of diffusion-based cross-attention maps to guide primitive placement, but the exact process for deriving these attention maps could be explained in more detail. For example, it’s mentioned that highlighted words are used to create cross-attention maps, presumably by leveraging a diffusion model’s internal attention. Suggestion: Clarifying how these maps are obtained would help readers fully grasp the initialization process and take the guesswork out.

Scope of Explainability:
The paper claims explainable sketches, and indeed the use of primitives and the tracking of their evolution provides a form of explanation. One weakness is that the explanation is mostly post-hoc: one can interpret the final sketch by seeing which primitive corresponds to which part of the text. However, the model itself doesn’t produce a human-language explanation, it just yields an interpretable sketch. Suggestion: It might be worth discussing this distinction. Perhaps in future work, the authors could consider linking the primitive-based explanation with actual explanatory text (e.g., labeling primitives with the word they correspond to). Currently, the explainability is strong visually, but not evaluable in a textual sense. Again, this doesn’t need to be implemented now, but an acknowledgment of what “explanation” means in this context (visual correspondence rather than a textual rationale) would clarify the contribution for readers and position the work within explainableAI more explicitly.

---

### Official Review · Reviewer_xR8B · 2025-03-22
**Neat idea and nice results**

**Rating:** 4
**Confidence:** 3

**Review:**

This paper presents an interesting investigation to visualize CLIP text embedding with primitive shapes, thereby constructing a novel text-to-sketch pipeline. In general, I find the idea is neat and the experiment results are promising. Therefore, I recommend acceptance.

**Pros**
- The paper is easy to follow and important technical details are provided.
- The use of primitive shapes makes the method highly interpretable and mathematically tractable.
- PLD is a notable contribution as it ensures that each primitive contributes meaningfully to the sketch, resulting in cleaner outputs and faster convergence.

**Cons**
- The application scenario for the proposed model is somewhat unclear because the paper does not explicitly define practical use cases where the proposed method can excel beyond academic or experimental purposes. The artifact is unseen in human arts. It neither produced realistic sketches nor simplistic sketches.
- The core idea of using basic primitives for sketch synthesis seems incremental when compared to prior works like CLIPDraw. The novelty lies more in optimization and implementation improvements rather than a groundbreaking conceptual advance.

---

### Official Review · Reviewer_5hde · 2025-03-25
**This paper introduces CLIPDraw++, a method for text-to-sketch synthesis using simple geometric primitives. I find the approach to be sufficiently novel, and the paper is supported by solid experiments and analysis.**

**Rating:** 4
**Confidence:** 3

**Review:**

This paper presents CLIPDraw++, a method for text-to-sketch synthesis that replaces complex stroke representations (like Bézier curves) with simple geometric primitives—lines, circles, and semicircles. The key idea is to model sketch generation as a set of linear transformations on these primitives, guided by CLIP embeddings and initialized using diffusion model attention maps. The authors introduce optimized initialization methods and a primitive-level dropout mechanism to reduce noise, and the method achieves strong performance both quantitatively and qualitatively. It’s a lightweight, training-free approach that produces clean, interpretable, and semantically aligned sketches.

Pros
Simplicity: Using basic primitives with linear transformations is a good innovation for generating sketches. It’s interpretable and easy to debug.
Good ablation and analysis: The paper includes useful comparisons against five existing methods—CLIPDraw, CLIPasso, CLIPascene, VectorFusion, and SVGDreamer—covering both optimization-based and diffusion-based baselines. It evaluates performance using a diverse set of metrics: Similarity, PSNR, CLIP-T score, BLIP score, and Confusion score. In addition to quantitative results, the paper presents qualitative visual comparisons, primitive-level tracking, and detailed ablations on components like primitive-level dropout, patch-based vs. point-based initialization, and CLIP vs. diffusion attention maps, making a strong case for each design choice.

Cons
Lack of Precision in Mathematical Notation: The mathematical expressions lack clarity and consistency. For example, in Equation (1), the notations F, F_tilde, F with underscript are confusing, In Equation (2), symbols like T and I are used without explanation, and in Equation (3), the meaning of “!” is unexplained. Overall, the paper would benefit from a careful review to ensure all variables and symbols are properly introduced and used consistently.

Some language is unclear or misleading: There are a few places where the writing could be tightened up. For instance, Line 214 says the model acts as a proxy for the sketch, but it’s really the embedding that does—it’s a subtle but important distinction. Also, Equation (3) is introduced without any explanation, which makes the intent hard to follow. I’d recommend having a few colleagues or labmates read through the paper to help identify spots where the phrasing could be clearer.

Concerns with evaluation metrics: Using CLIP-T scores for evaluation raises concerns about circular evaluation—since the model is directly optimized for CLIP similarity, relying heavily on CLIP-based metrics may overstate its performance. Also, these metrics don’t always align with human perception. For example, in the "horse eating a cupcake" case, the result from SVGDreamer looks more realistic, yet gets a lower CLIP-T score. But I understand it is common practice in this field.

---

### Official Review · Reviewer_vHKk · 2025-03-25
**Interesting geometric primitive based sketch synthesis with initialization and optimization methods**

**Rating:** 4
**Confidence:** 3

**Review:**

This paper proposes CLIPDraw++, which contributes a simple geometric primitive-based approach for synthesizing sketches, which are more efficient and interpretable. The paper is easy to follow.

The main technical contributions are in canvas initialization and optimization. Compared to random initialization in ClipDraw, ClipDraw++ uses cross-attention maps from a pretrained diffusion model to identify important regions and initialize primitives in patches. The paper also demonstrates the effectiveness of primitive-level dropout that reduces noisy strokes and improves quality. The learnable opacity parameter takes inspiration from human sketching practices, and appears to be effective in controlling noise in the generated sketch.

My concern is with the reliance on CLIP model, which the method relies on and also is evaluated against as a metric. The quantitative comparison is also fairly limited in its details, such as the number of prompts/sketches generated. Moreover, while the method is mostly evaluated on single objects or simple scenes, its effectiveness for complex or multi-object scenes has not been thoroughly evaluated.

Overall, I feel CLIPDraw++ is unique in its insights on using simple geometric primitives and the initialization and optimization methods proposed and would be an interesting paper to include in the workshop.

---

### Decision · Program_Chairs · 2025-03-25

**Decision:**

Accept

**Comment:**

The paper introduces CLIPDraw++, an innovative method for text-to-sketch synthesis using simple geometric primitives guided by CLIP embeddings and diffusion-model attention maps. The reviewers highlighted the approach's simplicity, interpretability, and strong experimental validation, including detailed ablations and meaningful qualitative comparisons. Minor concerns include limited quantitative runtime evaluation, unclear mathematical notation, and challenges with scaling to complex scenes.

Given the positive feedback emphasizing novelty, thorough validation, and clear technical contributions, the paper is accepted. Authors are recommended to clarify mathematical notation, discuss limitations regarding complex prompts explicitly, and include detailed runtime analyses in the camera-ready version.